# Coastal retreat rates of high-Arctic rock cliffs on Brøgger peninsula, Svalbard, accelerate during the past decade

Juditha Aga[1], Livia Piermattei[1,2], Luc Girod[1], Kristoffer Aalstad[1], Trond Eiken[1], Andreas Kääb[1], and Sebastian Westermann[1,3]

[1]Department of Geosciences, University of Oslo, Norway.
[2]Swiss Federal Institute for Forest, Snow and Landscape Research (WSL), Birmensdorf, Switzerland.
[3]Center for Biogeochemistry in the Anthropocene, University of Oslo, Norway.

**Correspondence:** Juditha Aga (juditha.aga@geo.uio.no)

**Abstract.** In many Arctic regions, marine coastlines change rapidly in the currently warming climate. In contrast, coastal rock cliffs on Svalbard are considered to be comparably stable, based on previous studies that considered only a few years and limited coastal reaches. Long-term trends of coastal retreat rates in rock cliffs on Svalbard are unknown so far, but their quantification could improve the understanding of coastal dynamics on the Arctic archipelago. This study presents coastal retreat rates in rock cliffs along several kilometers of the Brøgger peninsula, Svalbard. The analysis relies on high-resolution orthoimages from 1970, 1990, 2010 and 2021, corroborated by high-precision dGNSS measurements along selected segments of the coastline and rock surface temperature measurements during the period 2020-2021. Our analysis reveals statistically significant acceleration in coastal retreat rates along the Brøgger peninsula between 2010 and 2021. This is true for both the northeast facing coastline, with retreat rates increasing from $0.04\pm0.06\,\mathrm{m\,a^{-1}}$ (1970-1990) and $0.04\pm0.04\,\mathrm{m\,a^{-1}}$ (1990-2010) to $0.06\pm0.08\,\mathrm{m\,a^{-1}}$ (2010-2021) and the southwest facing coastline, where retreat rates of $0.26\pm0.06\,\mathrm{m\,a^{-1}}$ (1970-1990), $0.24\pm0.04\,\mathrm{m\,a^{-1}}$ (1990-2010) and $0.30\pm0.08\,\mathrm{m\,a^{-1}}$ (2010-2021) are measured. This corresponds to an increase in the most recent decade of 75 % for the northeast facing coastline and 25 % for the southwest facing coastline. Furthermore, the parts of the coastline affected by erosion increase along the northeast facing coastline from 47 % (1970-1990) to 65 % (2010-2021), while they stay consistently above 90 % along the southwest facing coastline. Measurements of rock surface temperature show mean annual values close to the thaw threshold with -0.49°C at the southwest facing coastline, while records at the northeast facing coastline are lower with -1.64°C. The recently accelerated retreat rates coincide with increasing storminess and retreating sea ice, together with increasing ground temperatures, all factors that can enhance coastal erosion.

## 1 Introduction

Arctic permafrost environments can respond rapidly to changing climatic conditions (Biskaborn et al., 2019). This is especially true for Arctic coastlines where sea level rise, sea ice retreat, increasing seawater temperatures and loss of permafrost and glaciers have pronounced effects on coastal dynamics (Irrgang et al., 2022). Therefore, Arctic coasts are often eroding more rapidly than coasts in temperate regions and the average retreat rate is estimated to $0.5\,\mathrm{m\,a^{-1}}$ (Lantuit et al., 2012). However, the variability of coastal retreat rates across the Arctic is pronounced, both on a regional and local scale. Lantuit et al. (2012)

present a circum-Arctic database, where the highest rates are detected in the Laptev Sea ($0.73 \ \mathrm{m\,a^{-1}}$), the East Siberian Sea ($0.87 \ \mathrm{m\,a^{-1}}$), the US Beaufort Sea ($1.15 \ \mathrm{m\,a^{-1}}$) and the Canadian Beaufort Sea ($1.12 \ \mathrm{m\,a^{-1}}$). Numerous regional studies corroborate these numbers, for example with retreat rates along the Bykovsky peninsula (Laptev Sea) of $0.59 \ \mathrm{m\,a^{-1}}$ between 1951 and 2006 (Lantuit et al., 2011), along the US Beaufort Sea of $1.8 \ \mathrm{m\,a^{-1}}$ between 1940 and 2010 (Gibbs and Richmond, 2017) and along the Canadian Beaufort Sea with $0.7 \ \mathrm{m\,a^{-1}}$ (Irrgang et al., 2018). The highest erosion rates are often found in ice-rich permafrost bluffs and barrier islands. Jones et al. (2018) present a maximum of $48.8 \ \mathrm{m\,a^{-1}}$ in such a setting along the US Beaufort Sea from 2007 to 2008.

Following the changing climatic conditions, erosion rates along the Arctic coastline have increased since the early 2000s (Jones et al., 2020). This trend is expected to continue with a sensitivity to warming of 0.4 to $0.8 \ \mathrm{m\,a^{-1}\,{}^{\circ}C^{-1}}$ (Nielsen et al., 2022). Increasing coastal erosion rates can enhance organic carbon and sediment fluxes into the nearshore zone (Fritz et al., 2017; Tanski et al., 2017) and threaten infrastructure, settlements and archaeological sites (Radosavljevic et al., 2016).

In contrast, lithified coastlines in the Arctic are assumed to be relatively stable and show smaller erosion rates compared to unlithified Arctic coastlines. The database of Lantuit et al. (2012) presents retreat rates for the Canadian Archipelago of $0.01 \ \mathrm{m\,a^{-1}}$ and for Svalbard of $0.00 \ \mathrm{m\,a^{-1}}$. This can be explained by the higher resistance against erosion due to the lithology (Irrgang et al., 2022). Lithified coastlines are often dominant along Arctic archipelagos, such as the Canadian Arctic Archipelago, Greenland and Svalbard. Here, the topography is often characterized by fjords and narrow straits, impeding a long fetch and protecting the coastlines against wave activity (Overduin et al., 2014).

The coastal environment of Svalbard is highly affected by paraglacial processes caused by the glacial retreat since the Little Ice Age (Bourriquen et al., 2016). Consequences are the exposure of new coastal landscapes (Kavan and Strzelecki, 2023), an intensified glaciofluvial sediment transport towards the coast resulting in coastal progradation (Mercier and Laffly, 2005; Bourriquen et al., 2018), but also an increased exposure towards storms and wave action enhancing coastline retreat (Zagórski et al., 2015). In addition, terrestrial processes such as precipitation events can contribute to erosion, especially in unlithified material (Sessford et al., 2015). Regional studies on Svalbard show that the retreat of coastal rock cliffs is in the order of mm to cm per year: a yearly retreat of coastal cliffs by 2.7 to $3.1 \ \mathrm{mm}$ from 2002 to 2004 was observed in Kongsfjorden (Wangensteen et al., 2007) and by up to $1.9 \ \mathrm{cm}$ between 2014 to 2015 in Hornsund (Lim et al., 2020). In contrast, mixed-type coasts on Svalbard, i.e. bedrock cliffs covered by unconsolidated sediment, have been documented to show larger erosion rates of up to $80 \ \mathrm{cm}$ per year (Guégan and Christiansen, 2017).

Warming is especially intense on Svalbard, as the archipelago is influenced by the northern part of the warm North Atlantic current, making it more susceptible to atmospheric and oceanic changes (Walczowski and Piechura, 2011). Air temperatures have increased in the past decades (Nordli et al., 2020) and are projected to rise also in the future, with the highest rates in the winter season (Hanssen-Bauer et al., 2019; Isaksen et al., 2016). Following this atmospheric trend, borehole data from the last decades show increasing permafrost temperatures on Svalbard (Boike et al., 2018; Christiansen et al., 2010; Etzelmüller et al., 2020; Isaksen et al., 2007). The effect of increasing air temperature on permafrost degradation and thaw is expected to be especially pronounced along the coastline, as the ground can be warmed both from the top and laterally from the bluff face (Irrgang et al., 2022).

In addition to increasing air and ground temperatures, sea ice coverage has been markedly reduced around Svalbard. A
prime example is Kongsfjorden, which was typically covered by sea ice during the winter season (Gerland and Hall, 2006).
However, since 2002, the sea ice duration has shortened and after 2006, the maximum extent of sea ice was drastically reduced
(Johansson et al., 2020). Longer ice-free seasons enhance the influence of wave action and storms (Overeem et al., 2011).
Furthermore, radiative warming by the relatively warm seawater can lead to higher ground surface temperatures along the
coast (Schmidt et al., 2021). Both the exposure to waves and storms and the warming trend in permafrost could contribute to
an increased vulnerability of the coasts in Svalbard.

Previous studies on coastal cliff erosion on Svalbard cover only a few years (Guégan and Christiansen, 2017; Lim et al.,
2020; Prick, 2004; Wangensteen et al., 2007) and long-term analyses are lacking so far. These can be especially valuable, as
short-term studies may be biased by the presence or absence of single storm events that can contribute considerably to coastal
erosion. The present study focuses on the retreat rate of the sediment-covered coastal cliffs along the Brøgger peninsula,
Svalbard, based on high-resolution historical and digital aerial images (1970 to 2021). The main objectives of this study are (i)
to detect long-term trends in coastal retreat along the Brøgger peninsula, separately analyzed for the northeast and southwest
facing coastline, (ii) to analyze rock surface temperatures for both expositions of the coastline, and (iii) to link these changes
to available climate data.

## 2   Study area

The study area is the northwestern part of the Brøgger peninsula, located on the west coast of Spitsbergen. It stretches from
the southwest at approximately 78°55' N, 11°15' E to the northeast at 78°59' N, 11°40' E. It has a total coastline of about
14 km of which we analyzed approximately 5.5 km, where rock cliffs behind beach sediments show active coastal erosion of
the bedrock (Etzelmüller et al., 2003). The bedrock is typically highly fractured (Ødegård and Sollid, 1993) and dominated
by conglomerates, sandstones, shales and carbonates from the Carboniferous to Permian age, which often form overhanging
cliffs (Fig. 1). They are typically covered with several meters of unconsolidated sediments, consisting of raised beach ridges
(Etzelmüller and Sollid, 1991), which are dated to the Late Weichsel (about 13.5 ka) below 45 masl (Forman et al., 1987)
and uplifted following the isostatic rebound of the land caused by the retreat of glaciers (Rotem et al., 2023). The current
uplift rate in Ny-Ålesund is $8.0\pm0.3\ \mathrm{mm\,a}^{-1}$ (Hanssen-Bauer et al., 2019). As the unconsolidated deposits are more prone to
erosion, they often form a slightly retreated line above the bedrock (Fig. 1). The coastal cliffs have a mean height of 15.5 m
with a maximum of 28.0 m, whereof the bedrock accounts for approximately 10.5 m on average. The average slope angle of
the unconsolidated sediments is approximately 35°.

The field site is characterized by continuous permafrost, even though the presence of taliks cannot be excluded. At the
Bayelva soil and climate station, which is about 8 km distance to the investigated field site, mean annual ground temperatures
in a depth of 9 m are recorded with -3.0 to -2.6°C between 2009 and 2016 (GTN-P, 2018). Measurements of rock surface
temperatures in the coastal cliffs of Brøgger peninsula in about 8 km distance to the field site also revealed relatively warm
permafrost, with annual values between -0.6 and -3.6°C in the years 2016 to 2020 (Schmidt et al., 2021).

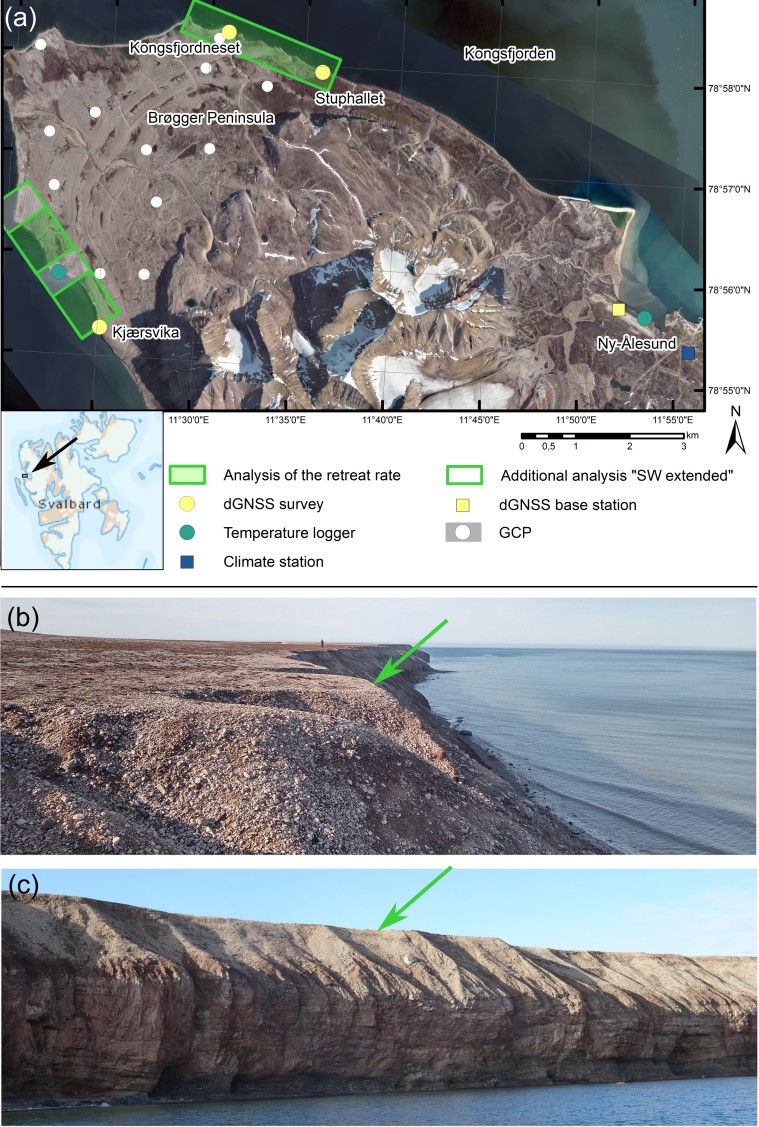

**Figure 1.** (a) Orthoimage of the study area, including the measurement sites. The cliff top retreat rate is analyzed along the northeast and southwest facing coastline with an extended part in the southwest. The locations of the dGNSS surveys and the corresponding base station, the temperature loggers, the climate station and the ground control points (GCP) are shown as dots. Source: NP_Basiskart_Svalbard_WMTS_32633 © Norwegian Polar Institute. The (b) top and the (c) bluff face along the coastline of Brøgger peninsula show the rock cliffs covered with unconsolidated sediments. The green arrows and lines indicate the top of the cliff that is digitized. Both photos were taken at the location of the dGNSS survey at Stuphallet.

The bedrock cliffs on Svalbard are exposed to several denudational processes. At the shore platform and the lower part of the coastal cliff, abrasion is likely the main controlling factor with wave action acting upon the bedrock (Are, 1988a, b),

redistributing beach sediments and consequently polishing the bedrock (Strzelecki et al., 2017). In addition, wetting-drying cycles by tidal water level changes and freeze-thaw processes can weaken the bedrock, especially where open cracks are present. In winter, an icefoot or snowdrift can develop, protecting the shore platform and the lower parts of the coastal rock wall from denudational processes. Above, where waves cannot reach the coastal cliff, periglacial weathering is controlling the erosion (Strzelecki et al., 2017). Here, rock fracturing through ice segregation may contribute to an increased susceptibility of the bedrock (Ødegård and Sollid, 1993). The unconsolidated sediments on top of the bedrock rest with their natural friction angle, following the erosion of the bedrock.

Meteorological data such as air temperature and solar radiation are measured by climate stations located in Ny-Ålesund (Fig. 1), which is approximately 8 km southeast of the study area and located at the northeast facing coastline of the Brøgger peninsula. Station data shows that air temperatures increased from an average of -5.9°C in the 1970s to -3.1°C in the 2010s (Norwegian Meteorological Institute, 2022b). This corresponds to a linear increase in mean annual air temperature for the period 1971 to 2017 of 0.71°C per decade, with the strongest increase in the winter season of 1.35°C per decade (Hanssen-Bauer et al., 2019). Maturilli et al. (2015), who looked at a shorter period from 1994 to 2013, detected an even stronger trend of 1.3±0.7°C per decade, with a winter warming of 3.1±2.6°C per decade. Hereby, the month with the lowest air temperatures is typically February, while the highest values are found in July (Hanssen-Bauer et al., 2019). The observed winter warming is associated with an increase in incoming longwave radiation of 15.6±11.6 $\mathrm{W\,m^{-2}}$ per decade. As the duration of snow cover is shortened and, hence, the reflection of shortwave radiation is reduced, the net shortwave radiation is slightly increased in the spring and summer seasons (Maturilli et al., 2015).

The mean annual precipitation between 2010 and 2021 in Ny-Ålesund was 526 mm, showing an increasing trend in the last decades since the 1980s with 384 mm (Norwegian Meteorological Institute, 2022a). Both snowfall and rainfall can occur at any time during the year, but the snow season typically lasts from October to June (Hop and Wiencke, 2019). Following the trend of warming air temperatures, a shifting of the onset of snow melt to earlier dates is observed with -5.8±8.3 days per decade (Maturilli et al., 2015). At the Bayelva soil and climate station typical snow depths between 0.65 and 1.4 m are observed (Boike et al., 2018). The steep coastal cliffs of Brøgger peninsula are typically snow-free. Here, snow accumulations are limited to edges in the bedrock and snow accumulations at the foot of the rock walls (Schmidt et al., 2021).

The wind regime of Brøgger peninsula is notably influenced by the mountainous terrain, the topography of Kongsfjorden and katabatic winds originating from the glaciers, resulting in a complex wind field (Svendsen et al., 2002; Maturilli and Kayser, 2017). In Ny-Ålesund, the occurrence of days, where mean hourly wind speeds with a strong breeze or stronger (wind speeds >= 10.8 $\mathrm{m\,s^{-1}}$) have been recorded, has increased from an average of around 65 days in the 1970s to 90 days in recent years (as described in Sect. 4.5; Norwegian Centre for Climate Services, 2023). It is important to note that the wind characteristics in Ny-Ålesund are not directly applicable to the study site because the influence of the mountains and the fjord diminishes and likely plays a lesser role. At the tip of Brøgger peninsula, which is part of the study site, wind speed measurements are available since the summer of 2021 (Norwegian Centre for Climate Services, 2023). These data reveal stronger wind speeds, with 112 days with strong breezes recorded in 2022, compared to 84 days in Ny-Ålesund, but long-term trends in wind speeds for the investigated field site are not available.

## 3    Data and methods

We used orthoimages derived from aerial nadir images acquired in 1970, 1990, 2010, and 2021 (Sect. 3.1) to analyze the retreat rate of the coastal cliffs along the coastline (Sect. 3.3). In August 2021, almost at the same time as the airborne image acquisition, we conducted a dGNSS survey along selected parts of the coastline (Sect. 3.2) to collect validation data. Rock surface temperatures were measured from September 2020 to August 2021 (Sect. 3.4). Furthermore, we used wind speeds records from Ny-Ålesund, sea ice charts and topographically downscaled ERA5 reanalysis data (Appendix A) to contextualize our results in light of ongoing climate change.

### 3.1    Orthoimages

The historical aerial images acquired in 1970 and 1990 were provided by the Norwegian Polar Institute (NPI). The 1970 data is composed of nine images (Table 1), which were taken with a RC8 camera (152 mm focal length) on an altitude of circa 2700 m in two flight lines. No calibration file was available to us and we used therefore the nominal calibration for this camera model. The grey-scale 1970 images were scanned from prints using an HP Expression XL10000 (A3 flatbed scanner), at 2400 dpi using 16-bit grey scale. The 31 images from 1990 (Table 1) were acquired with a RC20 camera (152 mm focal length) on an altitude of approximately 2700 m in seven flight lines. The calibration report was made available by NPI. The false-colour infrared images of 1990 were scanned by the Norwegian Polar Institute with a photogrammetric scanner at a resolution of 1800 dpi using 8-bit colour scale. The 1970 and 1990 aerial images were processed separately with MicMac, an open-source software for photogrammetry (Rupnik et al., 2017), with scripts available in Girod and Filhol (2022). The workflow identifies tie points to match the images and solves relative orientation, uses ground control points (GCPs) for bundle adjustment and refinement of the camera calibration, and finally generates a DEM and an orthoimage. We used the same GCPs for both the 1970 and 1990 images, which were identified and extracted from the 2010 DEM and orthoimage (Fig. 1). DEMs were produced for both 1970 and 1990. We estimated the a priori accuracy of the GCPs to be around 0.5 m, which was also confirmed by the results of the bundle adjustments. Both the 1970 and 1990 aerial photos were taken by a high-precision photogrammetric-class camera and multiple overlapping flight lines. The resulting DEMs showed no significant artefacts in comparison to the 2010 DEM. Nevertheless, the 1970 DEM exhibited more noise than the 1990 DEM. We thus decided to use the respective DEMs for the 1970 and 1990 orthoprojections.

The 2010 orthoimage and DEM were readily downloaded at a resolution of 0.20 m from the Norwegian Polar Institute base data of Svalbard (https://geodata.npolar.no/ and Norwegian Polar Institute (2014)). The 2021 aerial images were collected by the Svalbard Integrated Arctic Earth Observing System (SIOS) with a ground sampling distance of 0.08 m. High-precision onboard dGNSS and navigation data were available for these more recent images, enabling the use of geolocation tags from the metadata files to georeference the images. The accuracy of the georeferencing was confirmed by comparison with the 1970, 1990 and 2010 images. The orthoimages were also processed in MicMac (Girod and Filhol, 2022), but this time by mosaicking individually orthorectified images. The resolution of the final product was set to 0.20 m to ensure consistency with the resolution of the other images and to reduce the file size. The attributes of the orthoimages are reported in Table 1.

**Table 1.** Orthoimages used in this study and the respective metadata, the number and accuracy of the ground control points (GCP).

| Acquisition date | Image provider | No. of images | Orthoimage resolution [m] | No. of GCPs | Accuracy [m] of GCPs |
|---|---|---|---|---|---|
| 28 Aug 2021 | SIOS | 306 | 0.20 | 0 | N/A |
| 1 Aug 2010 | NPI | N/A | 0.20 | N/A | N/A |
| 14-22 Aug 1990 | NPI | 31 | 0.25 | 13 | 0.5 |
| 22 Aug 1970 | NPI | 9 | 0.20 | 13 | 0.5 |

### 3.2 dGNSS survey

The dGNSS measurements were used to validate if the manual digitization of the coastline from 2021 (Sect. 3.3) shows a systematic seaward or landward bias. They were well suited for that purpose as the dGNSS survey was conducted on August 31, 2021, while the aerial images were acquired on August 28, 2021. As no real-time connection to the base station in Ny-Ålesund could be established, logging of raw data in the field was necessary. Hereby, the receiver was placed in positions exactly at the cliff edge. dGNSS observations were made with an Altus APS3G GPS+Glonass receiver in a kinematic stop&go survey. The receiver was kept static at each point for 60 seconds with a 1-second log interval for raw data. Post-processing was made with the software RTK-lib ver. 3.4 (Takasu and Yasuda, 2009) relative to the permanent station NYA1 in Ny-Ålesund at a distance of 6 to 10 km.

The dGNSS surveys were conducted along three transects of the coastline of Brøgger peninsula: Kjærsvika, Kongsfjordneset, and Stuphallet (Fig. 1). The length of each transect is circa 40 m (10 points) at Kjærsvika, 275 m at Kongsfjordneset (45 points) and 182 m at Stuphallet (37 points). The number of points and the distance between them was adapted to the irregularity of the coastline, but on average we measured with intervals of 5 to 6 m. Standard deviations of dGNSS measured positions were a few cm and were considered error-free in the comparison with the digitized coastline, as the pixel size of the orthoimages are larger than the calculated errors.

### 3.3 Calculation of the cliff top retreat rates

The coastline was digitized along the top of the cliff (Fig. 1), which is slightly retreated compared to the actual shoreline, i.e. the boundary between water and land, due to unconsolidated sediments on top of the bedrock (Sect. 2). The top of the cliff has been used as a proxy for the shoreline in previous studies (e.g. Irrgang et al., 2018). However, it is important to note that the cliff top and the cliff foot can erode at different rates, and the presence of frontal beaches can affect the erosion processes (Swirad and Young, 2022). To address this, we conducted an analysis to confirm the suitability of the cliff top retreat as a proxy for coastal retreat at our field site. To do so, we compared the distance between the cliff top and the shoreline, as well as the width of the frontal beaches for 53 cross-sections along the coast with a proximate distance of 100 m. Hereby, we used the orthoimages from 2010 and 2021 since the shoreline could not be reliably detected in the orthoimages from 1990 and 1970.

The cliff top was digitized manually in a GIS environment using the WGS 1984 UTM Zone 33N at a scale of 1:400 by the same operator. The digitization process relied on visually interpreting the cliff top from the orthophoto, with additional visual

support from topographic maps, including hillshaded DEM and slope. Our analysis focused on the northeast and southwest facing coast of the Brøgger peninsula (Fig. 1) as the remaining coastline between Kongsfjordneset and Kvadehuksletta primarily consists of beaches, which are outside the scope of this study. The digitization along the bedrock coast was interrupted sporadically due to (1) rivers feeding into the fjord, which incised into the bedrock, (2) closely spaced thermo-erosional gullies, which prevented a clear detection of the coastline, and (3) the quality of the orthoimages. The last point affected only the digitization of the southwest facing coastline in the 1970 data, a challenging area due to unfavourable illumination conditions and excessive blurring of the orthoimage. In areas where the 1970 orthoimage met acceptable quality standards (i.e., exhibited less blur and higher contrast), we mapped the cliff top and the notches by tracing the boundary between the dark and light grey areas (as shown in Fig. 3). In this case, we assumed that the lighter area in the orthoimage corresponded to the steeper terrain of the cliffs. The hillshade and slope maps were very noisy and thus were not considered in the digitization process.

The retreat rate was calculated by comparing the cliff top position between the years 1970-1990, 1990-2010, and 2010-2021. Each cliff top position is subject to an uncertainty $U$, calculated following Irrgang et al. (2018) and Radosavljevic et al. (2016):

$$U = \sqrt{\mathrm{E_{GR}}^2 + \mathrm{RMS}^2 + \mathrm{LOA}^2} \qquad (1)$$

with $\mathrm{E_{GR}}$ being the ground resolution of the orthoimage, RMS representing the root mean square error created by the georeferencing of the aerial image, and LOA being the loss of accuracy due to digitizing errors. The RMS was estimated from manually measured offsets between the different orthoimages over stable terrain distributed over Kvadehuksletta (Fig. 1). The LOA was determined by repeating the digitization four times along the coastline segments where the dGNSS survey was conducted. Every meter along the three additional digitized lines, the distance to the original digitized coastline was calculated and the associated root mean square error served as the LOA for further uncertainty calculations.

The retreat rates were computed with the Digital Shoreline Analysis System (DSAS) version 5.1, an extension to Esri ArcGIS (Himmelstoss et al., 2021), which has been successfully applied in previous studies to detect changes in Arctic coastlines (e.g. Günther et al., 2013; Irrgang et al., 2018; Jones et al., 2018; Radosavljevic et al., 2016). Hereby, transects crossing all digitized coastlines are automatically generated with a user-defined spacing and the distance between these intersections is calculated. Due to the large number of transects along the coastline, statistical analyses of the cliff top retreat rates can be performed. We analyzed the periods 1970-1990, 1990-2010, and 2010-2021, constrained by the availability of orthoimages. As a first step, the distance between the digitized cliff tops was calculated separately for transects with a spacing of 5 m along the coastline (same spacing as in Radosavljevic et al. (2016)). Locally, irregularities of the coastlines may be distinct, e.g. protruding edges, and may lead to the case that a transect crosses the digitized coastline twice. Here, we always selected the smallest distance between two coastlines so as not to overestimate the retreat rates (Fig. 2). We calculated the retreat rate $RR$ for every transect by dividing the distance of the intersection $\Delta x$ (Fig. 2) by the analyzed time period $\Delta t$

$$RR = \frac{\Delta x}{\Delta t}, \qquad (2)$$

followed by the mean retreat rates as well as the percentage of transects that display a retreat.

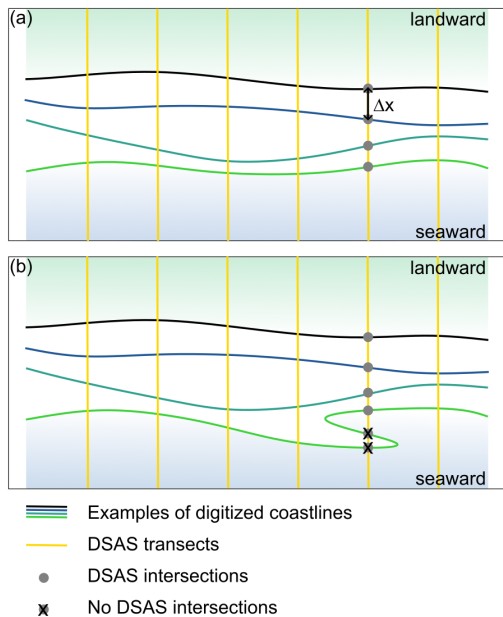

**Figure 2.** (a) Illustration of the analysis with DSAS, showing the digitized coastlines and the transects with intersections. The distance between two intersections is the retreat along one transect. (b) In the case of several intersections with one coastline, the smallest distance was used for analysis.

In total, 2.67 km of the northeast facing coastline and 1.73 km of the southwest facing coastline were analyzed for all orthoimages, in addition to 1.15 km only for the periods 1990-2010 and 2010-2021 along the southwest facing coastline, where the coastline in the orthoimage of 1970 could not be detected reliably. The uncertainty of the retreat rate, here dilution of accuracy (DOA), was calculated from the uncertainty of the cliff top position $U$ and the analyzed time span $\Delta t$ (Irrgang et al., 2018; Radosavljevic et al., 2016):

$$\text{DOA} = \frac{\sqrt{U_1^2 + U_2^2}}{\Delta t}. \tag{3}$$

To determine whether the increase (decrease) in the retreat rate is a statistically significant acceleration (deceleration) of the erosion, we performed a right-tailed (left-tailed) paired Student's $t$-test (Student, 1908) by comparing two consecutive retreat rates (e.g., 1970-1990 vs. 1990-2010) and evaluating the $p$-value using a threshold for significance of $\alpha = 0.005$ (Benjamin et al., 2018). $p$-values lower than this threshold can be considered statistically significant, which entails that the chance of observing a change in retreat as extreme as the one observed is extremely unlikely given the null hypothesis of no change (Ambaum, 2010; Benjamin et al., 2018).

### 3.4 Rock surface temperature monitoring

For analyzing the thermal regime of the rock walls, an iButton (© Maxim) temperature logger was installed in a rock wall close
to Kjærsvika, representing rock surface temperatures of the southwest facing coastline (RW-SW). No records could be gained
in the field area at the northeast facing coastline. Therefore, previously published rock surface temperature data were used in
this study (RW04), which were taken at an 8 km distance from the study area and followed the same installation technique as
described below (Schmidt et al., 2021). The naming of the temperature logger was adopted from the aforementioned study to
simplify the comparison. Including temperature data from these two settings allows the analysis of the thermal regime on both
expositions of the Brøgger peninsula (Fig. 1).

The measurements covered the time period from September 2020 to August 2021 with a sampling period of 4 hours. The
iButtons were placed in cracks in the coastal cliffs, ensuring direct thermal contact between the sensor with the rock surface
while protecting it from direct sunlight. The manufacturer states the measurement accuracy to 0.5°C. The numerical precision
of the sensor readout was set to 0.0625°C. We used no additional calibration. The uncertainty of the sensor and/or logger
system could be analyzed for the iButtons close to Ny-Ålesund, as several iButtons were placed in the same rock wall. Schmidt
et al. (2021) found uncertainties of less than 0.1°C for annual averages and less than 0.2°C for seasonal values. However, this
analysis was not possible for the iButton close to Kjærsvika as only a single sensor was operational during the study period.
The annual average for RW04 was calculated without a seven-day period in August 2021, for which no data are available.

### 3.5 Analysis of climate conditions

We analyzed trends in wind speed and changes in the distance to the sea ice edge (potential wave fetch), as these factors control
the interaction between wind and water and therefore the wave field (Barnhart et al., 2014), playing an important role along the
coastline of Brøgger peninsula due to mechanical abrasion through wave action. The wind speeds records were taken from the
Ny-Ålesund climate station SN99910 (78°55'23" N, 11°55'55" E; Fig. A1), covering the time period 1975 to 2020 (Norwegian
Centre for Climate Services, 2023). We extracted days during which mean hourly wind speeds of at least $10.8 \mathrm{~m\,s^{-1}}$ (strong
breeze or stronger) were recorded, corresponding to large waves of approximately 3 to 4 m (NTNU, 2023).

We also analyze the distance to the sea ice edge in northwesterly direction (corresponding to the open Fram Strait), which
is the potential distance over which waves can build up. The analysis of this potential fetch was based on data provided by the
Norwegian Meteorological Institute (2023) for the time period 1997 to 2023. Hereby, we define the sea ice edge as the given
category of 10 to 40 % sea ice concentration, following Meier and Stroeve (2008) and Overeem et al. (2011), who applied a
threshold of 15 %. We determined the mean distance to the sea ice edge for September for which 22 ice charts per year were
available on average. For trend detection, we applied a Bayesian regression analysis (Särkkä, 2013), which is explained in
Appendix A. In all other cardinal directions, land is found in about 10 to 15 km distance, so that the potential fetch is limited.

In addition, we used hourly ERA5 reanalysis data (Hersbach et al., 2020) in conjunction with the topography-based down-
scaling routing TopoSCALE (Fiddes and Gruber, 2014) to analyze trends in mean annual air temperature, annual rain- and

265 snowfall, as well as mean annual incoming longwave and shortwave radiation. The methods and detailed results for these climatic parameters are presented in Appendix A and Appendix B.

## 4 Results

### 4.1 Digitization of coastlines

Cliff top retreat rates were analyzed for three different time periods 1970-1990, 1990-2010, and 2010-2021, based on the
270 digitized coastlines on the respective orthoimages. Figure 3 shows a selected segment of the southwest facing coastline for all time periods, together with the digitized coastlines and the results of the dGNSS survey. The uncertainty of the digitized coastline is approximately 0.6 m for 1990, 2010 and 2021, while it is 1.07 m for the year 1970, based on multiple digitization of the coastline in selected transects (Sect. 3.3). However, as the analyzed time periods are one decade or more, the uncertainty for the retreat rates per year is reduced to 0.08 m or less (Table 2).

In addition, the digitized cliff top of 2021 was validated with the dGNSS measurements. The mean distance between the dGNSS points and the digitized cliff top was 0.33 m, which is approximately 1-2 pixels on the orthoimage (Table 1). The calculated bias is 0.12 m, indicating a slight landward shift of the digitized coastline. However, this is mainly attributable to four dGNSS points with a distance of more than 2 m, where protruding edges weren't detected in the orthoimage due to an unfavourable illumination. Removing those led to a mean distance between the dGNSS points and the digitized coastline of
0.23 m and a bias of 0.00 m. Consequently, typically no systematic seaward or landward bias could be detected, however, some protruding edges might not be captured with the digitized coastline.

**Table 2.** Uncertainty associated with the digitization of the coastline. $E_{GR}$ stands for the resolution of the orthoimage, RMS for the root mean square error associated with the georeferencing of the images and the noise from the processing, LOA for the loss of accuracy due to digitization, U for the uncertainty of the coastline position and DOA for the dilution of accuracy.

| Year of orthoimage | $E_{GR}$ [m] | RMS [m] | LOA [m] | U [m] | DOA [m] |
|---|---|---|---|---|---|
| 2021 | 0.20 | 0.5 | 0.23 | 0.59 | |
| 2010 | 0.20 | 0.5 | 0.28 | 0.61 | 0.08 |
| 1990 | 0.25 | 0.5 | 0.25 | 0.61 | 0.04 |
| 1970 | 0.20 | 1 | 0.33 | 1.07 | 0.06 |

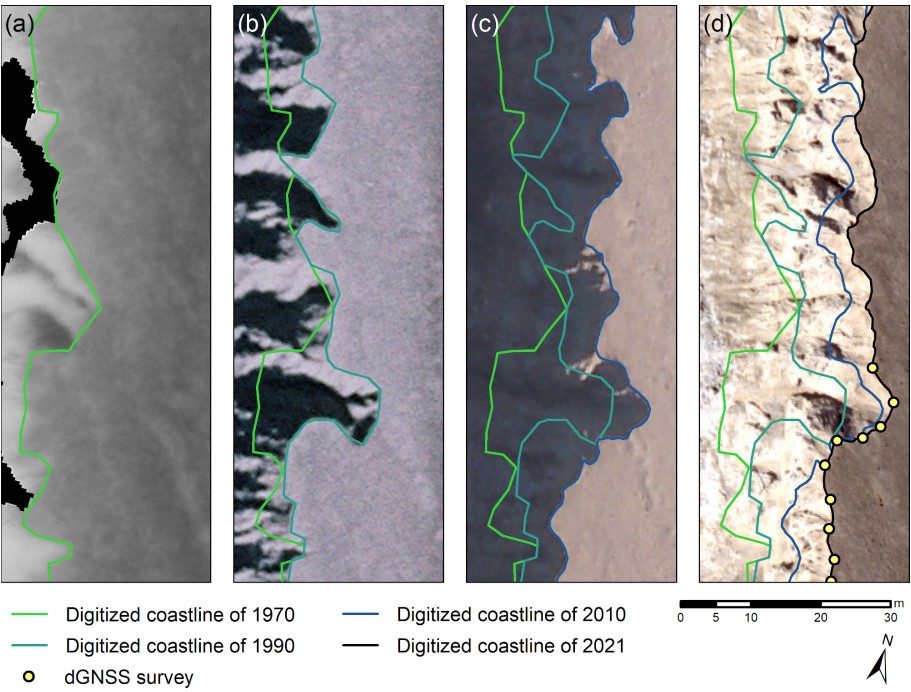

**Figure 3.** Orthoimages and digitized coastlines from (a) 1970, (b) 1990, (c) 2010, and (d) 2021 along a selected transect of the southwest facing coastline. The sea is to the left (west) of the image sections shown. The results of the dGNSS survey agree well with the digitized coastline of 2021. Source of the orthoimages: (a) and (b): Norwegian Polar Institute, not publicly available; (c): Norwegian Polar Institute, https://geodata.npolar.no/; (d): Svalbard Integrated Arctic Earth Observing Systems SIOS, not publicly available.

## 4.2 Cliff top retreat as a proxy for coastal retreat

The analysis of 53 cross-sections along the investigated coastline of Brøgger peninsula shows that the distance between the cliff top and the shoreline changed only slightly by less than 0.10 m between 2010 and 2021. An example of a representative cross-section is given in Fig. 4. Here, the distance from the cliff top to the shoreline was 32.35 m in 2010 and reduced marginally to 32.19 m in 2021. Meanwhile, the width of the beach increased slightly from 2.82 m in 2010 to 3.05 m in 2021. This example showcases that the cliff top retreats at similar rates as the shoreline.

The mean distance for all cross-sections was 17.44 m in 2010 and reduced marginally to 17.35 m in 2021. The change in distance of 0.09 m is considerably smaller than the uncertainty associated with the digitization of the coastline (Table 2), i.e. the position of the cliff top. Also, the distribution of the distances is comparable between those two years with clusters of cliff top-shoreline-distances around 4 m, 12 m and 24 m (Fig. 5).

In addition, we analyzed the width of the frontal beaches along the 53 cross-sections. The results show that the width was only slightly reduced from 2.23 m in 2010 to 2.09 m in 2021. Furthermore, the characteristics of the cliff morphology did not change significantly: Seawater reaching the cliff foot directly was detected for 40 % (2010) and 42 % (2021) of the cross-

295 sections, while 47 % (2010) and 45 % (2021) had frontal beaches with a high potential of inundation during stormy conditions (as the example given in Fig. 4), and only 7 % (2010, 2021) had extended frontal beaches, which could limit the effect of wave activity on the cliff foot. Both the small change in beach width and cliff morphology are indicators, that the eroded material is transported away effectively from the foot of the cliff and that no significant accumulation of eroded material occurs. Furthermore, we can conclude that the potential for wave activity is not affected due to changes in the cliff morphology

over the years.

As our results suggest that the average distance between the cliff top and the shoreline or the width of the beach does not change significantly over time, we are confident that the retreat of the cliff top is an applicable proxy for the coastal retreat at our field site.

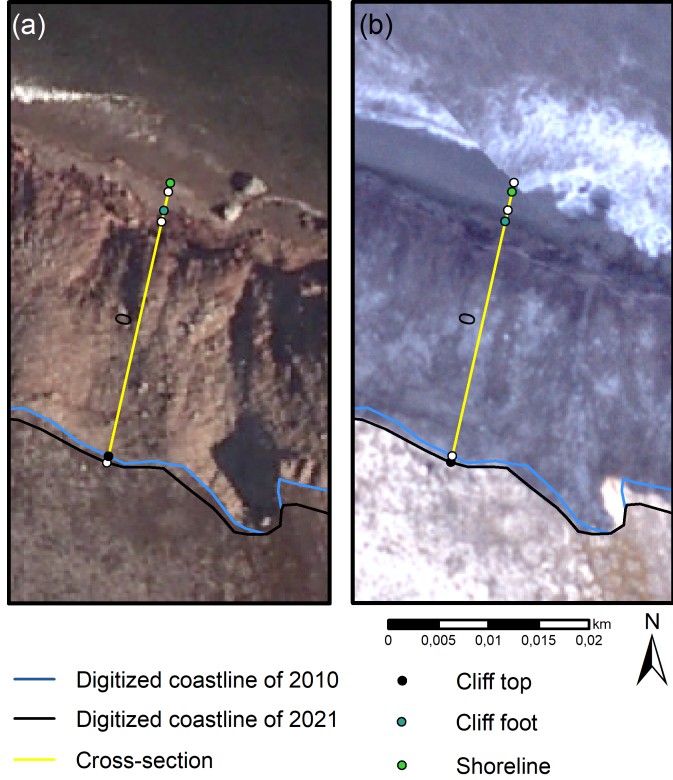

**Figure 4.** Cross-section with cliff top, cliff foot and shoreline for (a) 2010 and (b) 2021. The background shows the respective orthoimages. Source of the orthoimage: (a) Norwegian Polar Institute, https://geodata.npolar.no/; (b) Svalbard Integrated Arctic Earth Observing Systems SIOS, not publicly available.

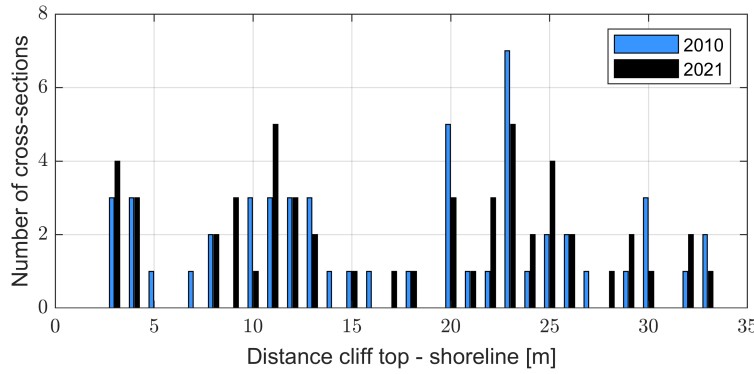

**Figure 5.** Number of cross-sections that show a certain distance between cliff top and shoreline for 2010 and 2021. The distribution only changes slightly with clusters around 4 m, 12 m and 24 m.

## 4.3 Coastal retreat rates on the Brøgger peninsula

The analysis for the different time periods was performed separately for the northeast and southwest facing coastline, due to markedly different retreat rates. The resulting retreat rates are visualized in Fig. 6 and all results (retreat rates and their uncertainties, percentage of transects calculating erosion and $p$-values for significant acceleration of the retreat rates) are summarized in Table 3.

**Table 3.** Mean retreat rates and associated statistics for the analyzed time periods. The analysis is performed separately for the northeast facing coastline, the southwest facing coastline and an extended part of the southwest facing coastline, where the coastline could not be detected in the image of 1970. T stands for transects, RR for retreat rate, DOA for dilution of accuracy, $T_E$ for transects calculating erosion and P is the result of Student's $t$-test. We tested for the significance of acceleration (right tailed), only the $p$-value denoted with $^\dagger$ is calculated for deceleration (left tailed).

| Extent | Length [m] | No. of T | Time period | RR ± DOA [m a$^{-1}$] | $T_E$ [%] | Max RR [m a$^{-1}$] | RR > 0.2 m a$^{-1}$ [%] | $p$-value ($t$-test) |
|---|---|---|---|---|---|---|---|---|
| NE | 2670 | 534 | 1970-1990 | 0.04 ± 0.06 | 47 | 0.51 | 2 | 0.409 |
| | | | 1990-2010 | 0.04 ± 0.04 | 55 | 0.44 | 1 | |
| | | | 2010-2021 | 0.06 ± 0.08 | 65 | 0.83 | 4 | 2.5E-10 |
| SW | 1730 | 346 | 1970-1990 | 0.26 ± 0.06 | 94 | 1.06 | 55 | 0.099$^\dagger$ |
| | | | 1990-2010 | 0.24 ± 0.04 | 92 | 1.28 | 52 | |
| | | | 2010-2021 | 0.30 ± 0.08 | 95 | 0.96 | 65 | 1.4E-6 |
| SW ext. | 2880 | 576 | 1970-1990 | - | - | - | - | |
| | | | 1990-2010 | 0.16 ± 0.04 | 81 | 1.28 | 32 | 0.003 |
| | | | 2010-2021 | 0.21 ± 0.08 | 89 | 0.96 | 45 | |

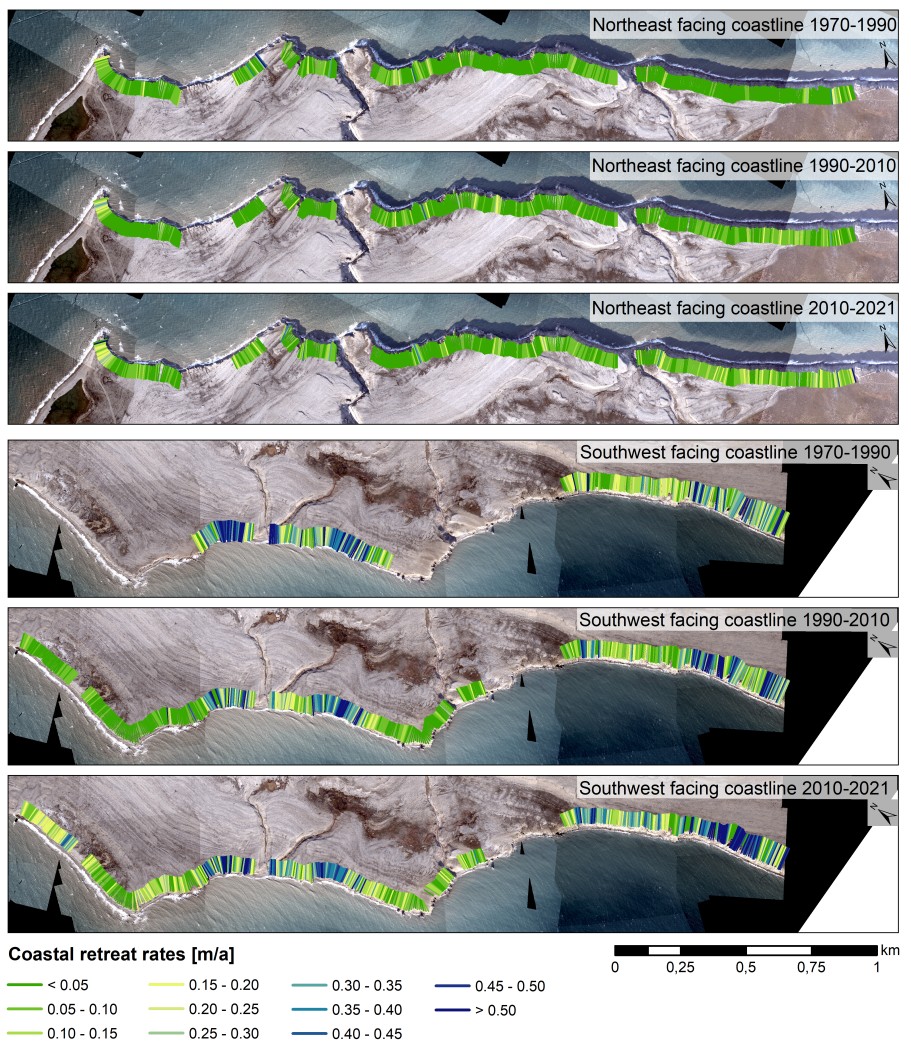

**Figure 6.** Cliff top retreat rates at the northeast and southwest facing coastline of Brøgger peninsula for the time periods 1970-1990, 1990-2010 and 2010-2021. The background shows the orthoimage of 2021. Source of the orthoimage: Svalbard Integrated Arctic Earth Observing Systems SIOS, not publicly available.

The northeast facing coastline can be considered fairly stable, but with an acceleration of erosion during the last decade. The time periods 1970-1990 and 1990-2010 show a retreat rate in the same order of magnitude, with a mean value of $0.04\pm0.06 \mathrm{~m\,a}^{-1}$ and $0.04\pm0.04 \mathrm{~m\,a}^{-1}$, respectively. However, the number of transects recording erosion increases slightly from 47 % to 55 %, indicating that only about half of the coastline is subject to erosion. Despite the low erosion on average, single transects can have up to $0.51 \mathrm{~m\,a}^{-1}$ (1970-1990) and $0.44 \mathrm{~m\,a}^{-1}$ (1990-2010), even though only 2 % (1970-1990) and 1 % (1990-2010) of the transects result in retreat rates of $0.20 \mathrm{~m\,a}^{-1}$ or more. The cliff top retreat rate increases in the following decade (2010-2021), indicating higher erosion, with a mean retreat rate of $0.06\pm0.08 \mathrm{~m\,a}^{-1}$. The acceleration in

retreat rate between 1990-2010 and 2010-2021 is statistically significant ($p = 2.5 * 10^{-10}$). A retreat is calculated for 65 % of the transects, showing that a longer part of the coastline experiences erosion during that time span. This is visualized in Fig. 6: while small retreat rates ($< 0.05 \, \mathrm{m\,a^{-1}}$, little to no erosion) dominate in 1970-1990, the number of transects with higher retreat rates ($< 0.20 \, \mathrm{m\,a^{-1}}$) increases along the entire coastline in 2010-2021. In addition, more transects show a retreat rate greater than $0.20 \, \mathrm{m\,a^{-1}}$, with a maximum of $0.83 \, \mathrm{m\,a^{-1}}$.

In contrast to retreat rates well below $0.10 \, \mathrm{m\,a^{-1}}$ in the northeast facing sector, the southwest facing coastline is subject to more pronounced erosion. The highest retreat rates in the last decade are recorded here. While erosion decelerates slightly from $0.26 \pm 0.06 \, \mathrm{m\,a^{-1}}$ in 1970-1990 to $0.24 \pm 0.04 \, \mathrm{m\,a^{-1}}$ in 1990-2010, it accelerates statistically significantly ($p = 1.4 * 10^{-6}$) to $0.30 \pm 0.08 \, \mathrm{m\,a^{-1}}$ in the past decade (2010-2021). Almost the entire southwest facing coastline is subject to erosion during the analyzed time period (1970-2021) with over 90 % of the transects showing a retreat. In addition to the more intense erosion compared to the northeast facing coastline, the maximum retreat rate is also higher in all analyzed time periods with $1.06 \, \mathrm{m\,a^{-1}}$ in 1970-1990, $1.28 \, \mathrm{m\,a^{-1}}$ in 1990-2010 and $0.96 \, \mathrm{m\,a^{-1}}$ in 2010-2021. Retreat rates of more than $0.20 \, \mathrm{m\,a^{-1}}$ are obtained for 55 % (1970-1990), 52 % (1990-2010) and 65 % (2010-2021) of the transects. They are often concentrated in certain parts of the coastline and shift within the analyzed time periods.

For the time periods 1990-2010 and 2010-2021 the retreat rate is estimated for a longer section (SW ext. in Table 3). Here the results show that the erosion accelerates significantly from $0.16 \pm 0.04 \, \mathrm{m\,a^{-1}}$ (1990-2010) to $0.21 \pm 0.08 \, \mathrm{m\,a^{-1}}$ (2010-2021). Furthermore, the percentage of transects that calculate retreat increases from 81 % to 89 %. These results indicate that lower retreat rates occur in this section, but they show the same trend of intensified erosion over the last decade 2010-2021.

### 4.4 Rock surface temperatures

The measurement period of the logger RW-SW close to Kjærsvika lasted from 1 September 2020 to 31 August 2021. During this period, the logger measured a mean annual rock surface temperature of -0.49°C. This indicates below-freezing temperatures, but average ground temperatures in the permafrost behind the rock wall are likely close to the freezing point. The mean seasonal values were -6.2°C for winter (December to February), -3.1°C for spring (March to May), 6.6°C for summer (June to August) and 0.6°C for fall (September to November). The lowest rock surface temperature at RW-SW was recorded on both 31 January and 16 March with a temperature of -14.7°C, while the highest rock surface temperature was found on 3 August with 18.7°C (Fig. 7). The daily variability in the rock surface temperature is more pronounced in the spring and summer seasons, showing high-frequency variability and large amplitudes. In the fall and winter seasons, although the daily variability also has high amplitudes, the frequencies of the temperature fluctuations decrease.

The logger RW-SW is located at the southwest facing coastline of Brøgger peninsula, exposed to the open sea. Comparing its records to the logger RW04 (Schmidt et al., 2021) at the northeast facing coastline, which is protected from the open sea by Kongsfjorden, shows differences in mean annual rock surface temperatures. RW04 measures a value of -1.64°C, which is 1.15°C lower than RW-SW. The temperature differences are particularly pronounced in late winter and early spring. The entire measurement period is shown in Fig. 7.

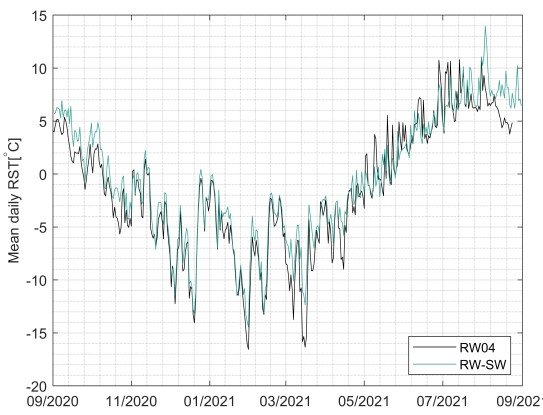

**Figure 7.** Measured rock surface temperature from September 1, 2020, to August 31, 2021 for both rock wall loggers at the northeast facing coast (RW04) and the southwest facing coast (RW-SW).

## 4.5 Changes in climatic conditions

Given the lithology along the coastline of Brøgger peninsula, mechanical abrasion through wave action is likely a dominant factor for erosion. Therefore, we focus in this section on the wind conditions and changes in the sea ice cover. Other factors, such as precipitation patterns, air temperature and radiation, are presented in Appendix B.

The trend analysis of wind speeds in Ny-Ålesund defined as a strong breeze or stronger (wind speeds >= 10.8 m s$^{-1}$; corresponding to large waves of approximately 3 to 4 m; NTNU, 2023) shows an increase by 5.4 days per decade from 355 approximately 65 days on average in the 1970s to 90 days in the last decade (Fig. 8a). However, due to the strong variability, the evidence in favor of a trend remains weak (Bayes factor < 0.5, Appendix A). Years with exceptionally many days of high wind speeds occurred in the early 1990s, while they decreased slightly in recent years. We emphasize that the climate station providing the data, is located approximately 8 km away from the field area inside Kongsfjorden, where wind speeds are lower compared to the investigated field site. Wind speed measurements at the tip of Brøgger peninsula (available only since 2021) 360 recorded 112 days with strong breezes in 2022 compared to 84 days in Ny-Ålesund (Norwegian Centre for Climate Services, 2023), thus highlighting the potential differences in wind regimes.

In most cardinal directions, land is found in about 10 to 15 km distance, so that the potential wave fetch over open water is limited here. However, in northwestern direction, the coastline of Brøgger peninsula is exposed to the open sea towards Fram Strait and the potential wave fetch is limited by the distance to the sea ice edge, which is displayed in Fig. 8b for the month 365 of September. We detected an increasing trend from about 150 km to 250 km between 1997 and 2023, which accounts for approximately 39 km per decade (Bayes factor 1-2, strong evidence in favor of the trend, Appendix A). The largest values with a mean distance to the sea ice edge of more than 300 km were observed in 2020 and 2021.

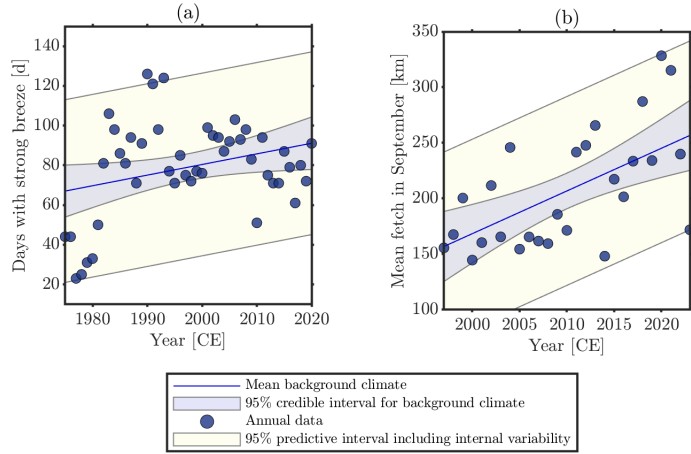

**Figure 8.** (a) Days with a strong breeze (wind speeds $>= 10.8\,\mathrm{m\,s^{-1}}$) measured by the Ny-Ålesund climate station SN99910. They increased by 5.4 days per decade between 1975 and 2020. (b) Mean distance to the sea ice edge (potential wave fetch) in September in northwesterly direction. The distance to the sea ice edge increased by approximately 39 km per decade between 1997 and 2023.

## 5 Discussion

### 5.1 Comparison to previous studies

In this study, we analyzed the retreat of the cliff top along the lithified coast of Brøgger peninsula. We showed that this rate is representative of the coastal retreat (Sect. 4.2) so that a comparison with the erosion along other coastlines on Svalbard and in the Arctic can be drawn.

The retreat rates indicate that the northeast facing coastline is fairly stable with mean annual values between $0.04\pm0.06\,\mathrm{m\,a^{-1}}$ and $0.06\pm0.08\,\mathrm{m\,a^{-1}}$ during the analyzed time period, while the southwest facing coastline is subject to more intense erosion with mean annual values between $0.24\pm0.04\,\mathrm{m\,a^{-1}}$ and $0.30\pm0.08\,\mathrm{m\,a^{-1}}$. Both cases show a statistically significant acceleration of erosion during the last decade 2010-2021. The calculated retreat rates are lower than the average change in Arctic coastlines of $0.5\,\mathrm{m\,a^{-1}}$ (Lantuit et al., 2012). This is expected as high retreat rates are typically found along unlithified coasts, which account for 65 % of the Arctic coastline (Irrgang et al., 2022). In contrast, the coastline along Brøgger peninsula is formed by bedrock, being characterized by a higher resistance against mechanical abrasion compared to unlithified coasts, and the unconsolidated sediments on top are not exposed to wave action. However, we detected higher retreat rates compared to other lithified coasts in the Arctic, e.g. the Canadian Archipelago with $0.01\,\mathrm{m\,a^{-1}}$ (Lantuit et al., 2012). This could potentially be explained by the long open water season at the western coast of Svalbard (Sect. 5.2), resulting in the high importance of mechanical abrasion by wave action (Sect. 5.2). Other contributing factors might be the highly fractured bedrock, decreasing the resistance of the material towards erosion and the permafrost conditions, which show a temperature range with decreased bedrock stability (Sect. 5.2).

In addition, the findings of this study suggest higher values than most previous studies on coastal cliff erosion on Svalbard. Prick (2004) reports retreat rates of $0.0019\ \mathrm{m\,a^{-1}}$ in coastal sandstone and shale cliffs in Longyearbyen using sediment traps and Wangensteen et al. (2007) detect a retreat of $0.0027\ \mathrm{m\,a^{-1}}$ and $0.0031\ \mathrm{m\,a^{-1}}$ with terrestrial photogrammetry in dolomitic limestone in Kongsfjorden area. Lim et al. (2020) find a higher retreat rate between $0.01\ \mathrm{m\,a^{-1}}$ and $0.019\ \mathrm{m\,a^{-1}}$ in marbles at

Veslebogen, applying terrestrial laser scanning and photogrammetry. In contrast, the results of Guégan and Christiansen (2017) along bedrock overlain by unconsolidated sediments show retreat rates of $0.45\ \mathrm{m\,a^{-1}}$ to $0.80\ \mathrm{m\,a^{-1}}$ along a 100 m transect, which is larger than the mean but close to the maximum retreat rates found in this study. Differences in the retreat rates can be explained by various factors, such as (1) the applied method and considered time period, (2) lithology, and (3) wave activity as well as local climate conditions. However, the respective importance of each factor is difficult to determine.

(1) Previous studies of coastal cliff erosion on Svalbard determined the retreat rate with either terrestrial photogrammetry or laser scanning (Wangensteen et al., 2007; Lim et al., 2020). By doing so, volume erosion at the cliff face is detected by locating e.g. single rock falls. These methods are typically constrained in space in time due to narrow beaches and a lack of historical data, so the results are highly dependent on the selected location and year of measurement. In contrast to that, we only consider the retreat rate at the top of the cliff, but extend the analysis over a much longer segment of the coastline (5.5 km) and a longer

time period (51 years). By choosing a close spacing of 5 m of the transects, we ensure to capture small irregularities along the coastline. Consequently, we measure a different parameter and over different scales in time and space compared to previous studies, which may result in different retreat rates.

  (2) Varying bedrock characteristics can lead to large differences in retreat rates. Global averages for erosion rates for limestone are typically within the range of 0.001 to $0.01\ \mathrm{m\,a^{-1}}$ and for shale around $0.01\ \mathrm{m\,a^{-1}}$ (Sunamura, 1992). The results

of this study show much higher retreat rates, which could be explained by the coastal environment and associated abrasion together with the removal of the eroded material at the base of the cliff. While the spatially constrained methods of previous studies on Svalbard can describe the bedrock type in more detail and are often limited to one rock type, the coastal cliffs in this study consist of conglomerates, sandstones, shales, and carbonates. A detailed characterization of the lithology along the coastline of Brøgger peninsula would enable a better comparison with previous studies. However, such a mapping is beyond

the scope of this study.

  (3) The local setting plays an important role in coastal cliff erosion. The wave activity may differ markedly between field sites, depending on the exposure to the open ocean and the fetch, as well as the duration of sea ice cover and the development of an ice-foot (Irrgang et al., 2022; Wangensteen et al., 2007). Furthermore, variations in climate variables such as air temperatures and radiation can influence the thermal regime of the bedrock and therefore its susceptibility towards erosion (Krautblatter et al.,

2013). As all these factors can vary on a small scale across Svalbard (Hanssen-Bauer et al., 2019), previous studies are well expected to result in different retreat rates.

## 5.2   Coastal retreat rates under a warming climate

Retreat rates of Arctic coastlines are governed by various drivers, dependent on the local coastal setting and environmental conditions (Irrgang et al., 2022). The lower part of the coastal cliffs is prone to abrasion, acting through the thermal and wave-

420 driven mechanical energy of the sea (Are, 1988a, b) and intensive wetting-drying during open-water season (Strzelecki et al., 2017). We assume that these factors play an important role in coastal erosion along Brøgger peninsula, as overhanging rock walls with a retreated foot of the cliff can be observed (Fig. 1c). Abrasion is especially effective during stormy weather, which likely intensified in the area around Brøgger peninsula during the past decades, showing a positive trend of days with a strong breeze in the last decades (Fig. 8a). Furthermore, extreme cyclone events regularly occur in the Arctic North Atlantic, with 20

to 40 events during winter. In Ny-Ålesund, an increasing trend of six cyclones per decade was detected from 1979-2015, which can be related to a decreasing sea ice extent in the region and large-scale atmospheric circulation changes (Rinke et al., 2017). Single weather events like this can support landsliding and consequently, they can have a localized but pronounced influence on the retreat rates.

However, the impact of windiness and wind-induced wave action is expected to vary along the coastline of Brøgger penin-

430 sula. The northeast facing coastline is characterized by a relatively sheltered position within the Kongsfjorden system, with land in most cardinal directions within a range of 10 to 15 km. This likely restricts the fetch and consequently wave activity, which may explain the lower coastal retreat rates in this sector. In contrast, the southwest to west facing coastline is more exposed to the open sea, especially in westerly and northwesterly directions in which the potential wave fetch is controlled by the distance to the sea ice edge in Fram Strait, which has clearly increased in the last decades (Sect. 4.5). Previous studies have

435 shown that an increasing fetch results in wave growth (Casas-Prat and Wang, 2020b), increasing the capability for wave-driven erosion (Casas-Prat and Wang, 2020a). Therefore, the increasing distance to the sea ice edge towards the open Fram Strait likely increases the wave activity and thus mechanical abrasion along southwest sector of Brøgger peninsula, likely explaining the higher erosion rates found here.

In addition to the large-scale sea ice conditions in Fram Strait, local sea ice coverage around the Brøgger peninsula can

influence the thermal regime of the rock walls (Schmidt et al., 2021) and potentially also wave action and mechanical abrasion (Barnhart et al., 2014). Dahlke et al. (2020) provides an overview of sea ice extent around Svalbard from 1980 to 2016. The results show a considerable decrease in sea ice coverage during winter and spring in Forlandsundet (where the southwest facing coastline is located), decreasing from 50-70 % until the early 2000s to below 10 % in recent years. Kongsfjorden (where the northeast facing coastline is located) experienced an increase in sea ice extent from around 40 % to 60 % in the 1990s and a

subsequent decrease to around 10 %. However, as our field area is located in the outer parts of Kongsfjorden (NE sector) and near the open Arctic ocean (SW sector) where typically less sea ice develops, even lower percentages of sea ice coverage are likely, with mostly ice-free conditions in the last decade.

The increasing influence of wave activity due to intensified storminess and sea ice retreat may to some extent be counteracted by a lowering of relative sea level by -4.5±0.4 $\mathrm{mm\,a^{-1}}$ in Ny-Ålesund (Hanssen-Bauer et al., 2019). On Svalbard, the global

mean sea level rise, caused mainly by changes in seawater density and land ice mass (Slangen et al., 2017), is outpaced by the isostatic uplift of the land and changes in the gravitational field following deglaciation (Slangen et al., 2017; Kavan and Strzelecki, 2023). This phenomenon is dynamic in space and time (Hanssen-Bauer et al., 2019) and due to its heterogeneity, the effect on the coastal cliff erosion is difficult to quantify (Luetzenburg et al., 2023). However, we assume that the influence of the increasing wave activity on coastal cliff erosion is stronger than the relative sea level lowering.

Under ongoing atmospheric warming in the last decades, increasing ground temperatures have been measured on Svalbard (Boike et al., 2018; Christiansen et al., 2010; Etzelmüller et al., 2020; Isaksen et al., 2007). Furthermore, retreating sea ice coverage and a consequently ice-free water body can warm coastal rock walls through additional longwave energy input (Schmidt et al., 2021). Higher ground temperatures can affect the stability of permafrost rock walls (Krautblatter et al., 2013). Rock surface measurements in 2020/2021 along the coastline of Brøgger peninsula show that the permafrost is close to the thaw threshold, with especially high values at the southwest facing coastline, falling into a temperature range with decreased stability (Davies et al., 2001). Due to limited snow accumulation in the steep coastal cliffs, the rock surface temperature is mainly linked to air temperature as well as longwave and shortwave radiation. Topographically downscaled ERA5 climate data (Fiddes and Gruber, 2014; Hersbach et al., 2020; Renette et al., 2023) for Brøgger peninsula show that air temperature and longwave radiation increased over the past decades, while shortwave radiation decreased slightly (Appendix B). The latter is much stronger at the southwest facing coastline (Appendix B), contributing to higher rock surface temperatures at logger RW-SW. If this trend continues, we expect a further increase in rock temperatures in the coastal cliffs of Brøgger peninsula, potentially making the rock more susceptible to erosion.

The coastal cliffs stabilize the unconsolidated material on top. Analysis of the orthoimages shows that the top of the cliff retreats typically in conjunction with the bedrock and consequently, the erosion of the sediments is highly dependent on the retreat of the bedrock below. Typically, retreat rates for single transects are in the same order of magnitude as the mean retreat rates. However, high retreat rates can occur locally and within only one time span. These values might be caused by large blocks being released in single events, which in return affect greatly the erosion of the overlying sediments and thus the cliff top. While the erosion of the bedrock is likely the most important controlling factor, more annual rainfall (Appendix B) can result in increased overland flow, forming channels on the surface and erosional gullies (Jorgenson and Osterkamp, 2005), which enhance the mechanical erosion of the sediments. The development of these features can occur either rapidly or evolve over longer time periods (Fortier et al., 2007; Godin and Fortier, 2012). Observations along the coastline in the field show locally the occurrence of such channels and the analysis of the orthoimages reveals that the gullies develop over several decades. The influence of rain events is expected to increase, as the dominance of rainfall over snowfall during Arctic summer and autumn is projected to occur in the 21st century (McCrystall et al., 2021).

Although the data in this study do not allow us to relate the retreat rates directly to records of climate stations or downscaled climate data, we can observe that the acceleration of the retreat rate coincides with increasing storminess and retreating sea ice, which may enhance coastal erosion, supported by increasing ground temperatures, which may lead to an increased susceptibility of the coastline.

With no infrastructure along the investigated coastline of Brøgger peninsula, the consequences for the population are expected to be minor. Widening the perspective on Svalbard, settlements and infrastructure are concentrated along lithified and unlithified coastlines, such as the major administrative center Longyearbyen. Jaskólski et al. (2018) identified ports, quays and structures for coastal protection as the most endangered infrastructure along the coast in Longyearbyen, where not only coastal retreat but also river erosion, permafrost thaw and solifluction play an important role. In addition, the cultural heritage of Svalbard is severely affected by coastal erosion (Nicu et al., 2020, 2021). Pan-Arctic, increased coastal erosion threatens

Arctic communities, e.g. in Alaska (Bronen and Chapin III, 2013; Brady and Leichenko, 2020) and Canada (Andrachuk and Pearce, 2010; Radosavljevic et al., 2016; Irrgang et al., 2019), and even forced relocation of some settlements (Bronen and Chapin III, 2013). As the results from our study show, this may not only concern unlithified, ice-rich coasts, which are known to be highly affected by coastal erosion (Irrgang et al., 2022), but also lithified coasts in the Arctic.

## 6  Conclusions

In this study, we calculate retreat rates of the coastal cliffs along Brøgger peninsula, Svalbard, based on aerial orthoimages from the years 1970, 1990, 2010 and 2021. While previous studies on coastal erosion on Svalbard are spatially and temporally limited, we present long-term trends in retreat rates, covering several kilometers along Brøgger peninsula. The main conclusions are as follows:

– At the northeast facing coastline, we detect fairly stable conditions, however with an increasing trend in the retreat rate from $0.04\pm0.06\,\mathrm{m\,a^{-1}}$ (1970-1990) and $0.04\pm0.04\,\mathrm{m\,a^{-1}}$ (1990-2010) to $0.06\pm0.08\,\mathrm{m\,a^{-1}}$ (2010-2021). About half of the transects show erosion, increasing from 47 % (1970-1990) over 55 % (1990-2010) to 65 % (2010-2021).

– In contrast, the southwest facing coastline shows higher retreat rates with $0.26\pm0.06\,\mathrm{m\,a^{-1}}$ from 1970 to 1990, a decrease to $0.24\pm0.04\,\mathrm{m\,a^{-1}}$ in 1990-2010 and statistically significant acceleration in retreat rates to $0.30\pm0.08\,\mathrm{m\,a^{-1}}$ in the past decade from 2010 to 2021. More than 90 % of the southwest facing coastline is affected by erosion during the entire time period.

– Installed temperature loggers recorded rock surface temperatures in the time period September 2020 to August 2021. Permafrost conditions are close to the thaw threshold with a mean annual rock surface temperature of -0.49°C at the southwest facing coastline, while the mean annual rock surface temperature along the northeast facing coastline in Ny-Ålesund is lower with -1.64°C.

– The coastal cliffs of Brøgger peninsula are exposed to changing climatic conditions, in particular, increasing storminess and retreating sea ice, but also warming permafrost and increasing rainfall. These factors likely contribute to the accelerating coastal erosion measured over the past decade.

**Appendix A: Topography-based downscaling of atmospheric reanalysis data and climate change detection**

To investigate links between the coastal erosion dynamics at the Brøgger peninsula and possible changes in the local climate,
we used hourly ERA5 reanalysis data (Hersbach et al., 2020) in conjunction with a downscaling routine. The ERA5 reanalysis
data can relatively accurately represent the mesoscale (scales 10-100 km) behavior of the atmosphere and the underlying
surface. To take into account hillslope scale effects we used the topography-based downscaling routine TopoSCALE (Fiddes
and Gruber, 2014). This scheme has been widely used for downscaling reanalysis data in complex terrain (e.g. Renette et al.,
2023). TopoSCALE uses several terrain parameters (elevation, slope, aspect, sky view factor, and horizon angles) to adjust
coarse scale meteorological forcing to a finer grid that can represent local topography. We analyzed trends in mean annual air
temperature, annual rain- and snowfall, as well as mean annual incoming longwave and shortwave radiation.

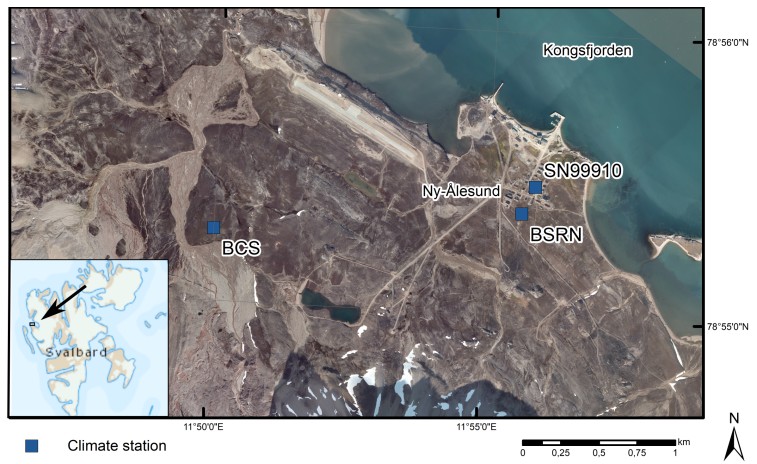

**Figure A1.** (a) Orthoimage of the surroundings of Ny-Ålesund, where the climate stations are located, which were used for validation of the
downscaled atmospheric data. Source: NP_Basiskart_Svalbard_WMTS_32633 © Norwegian Polar Institute.

The terrain parameters were computed using a 30 m resolution DEM of Svalbard (Norwegian Polar Institute, 2014), focusing
on 6 sites: the Ny-Ålesund climate station SN99910 (78°55'23" N, 11°55'55" E), the Ny-Ålesund BSRN station (78°55'22" N,
11°55'38" E), the Bayelva soil and climate station BCS (78°55'15" N, 11°50'0" E) (Fig. A1), Stuphallet (78°57'57" N,
11°35'57" E), Kongsfjordneset (78°58'21" N, 11°30'45" E), and Kjærsvika (78°55'17" N, 11°25'11" E) (Fig. 1). While the
three latter sites represent coastal cliffs, the first three were used for validation as shown in Fig. A2. The downscaling performed
well at a monthly time-scale for air temperature (validated against SN99910), specific humidity with a slight tendency to
underestimate periods with higher humidity (validated against BCS), incoming shortwave and longwave radiation (validated
against BCS) and provided a very accurate estimate of surface pressure (validated against BSRN). Although we did not perform
a direct validation of the downscaled precipitation, due to challenges in measuring this variable, the performance for surface
pressure suggests that ERA5 can capture larger scale synoptic precipitation events. The evaluation of wind speed showed a

larger scatter, so that station data was used when looking into trends in storminess (Sect. 3.5), provided by the Norwegian Centre for Climate Services (2023).

We aggregated the hourly downscaled data to represent annual means (temperature, incoming longwave, incoming short-wave) and accumulations (snowfall and rainfall). Annual snowfall and rainfall were estimated using an air temperature-based phase delineation of hourly precipitation. The annual aggregation leads to a further reduction in RMSE compared to the monthly estimates given in Fig. A2. The annual error is thus likely low enough for detecting multi-decadal climate change signals. For all variables but incoming shortwave radiation and days with storminess we take the average across the three study sites (Stuphallet, Kjærsvika, and Kongsfjordneset) for the downscaled ERA5 data from 1950 to 2021. For incoming shortwave radiation we use the same data and period but consider two sites with different exposition (Stuphallet: north east facing, Kjærsvika: south west facing) separately. For days with storminess, we used data from the Ny-Ålesund climate station SN99910 (Norwegian Centre for Climate Services, 2023).

To detect trends in the annual time series, we performed a Bayesian regression analysis (Särkkä, 2013) which is an established method for climate change detection (Annan, 2010). In particular, we fit two competing models: a "trend" model with a linearly varying background climate and a "steady" model with a fixed background climate.

The regression analysis of the trend model delivers the trend coefficients $\beta$ and the scales of internal climate variability $\sigma$. As a verification, we expect around 5 % of the data to fall outside the yellow 95 % predictive interval, which is what we see for all panels in Fig. B1. For each variable, we also report the Bayes factor (Kass and Raftery, 1995), denoted $B$, which quantifies how much support the data lends to the trend model versus the steady model. This is output on a logarithmic scale measuring the weight of evidence in favor of a trend: $0 < \log_{10}(B) < 0.5$ is *weak* evidence, $0.5 < \log_{10}(B) < 1$ is *substantial* evidence, $1 < \log_{10}(B) < 2$ is *strong* evidence, and $\log_{10}(B) > 2$ is *decisive* evidence. For negative $\log_{10}(B)$ the scale is the same but in favor of no trend (the steady model). For our analysis, $\log_{10}(B)$ can thus help quantify the strength of evidence in favor of (or against if $\log_{10}(B) < 0$) a climate change signal over the last 70 years.

## Appendix B: Trends in climatic parameters for the Brøgger Peninsula

The outputs from the regression analysis for the trend model are shown in Fig. B1. The air temperature shows a decisive positive trend with +0.6°C per decade, increasing from approximately -7°C to -3°C. A strong increase in annual rainfall is detected with 13 mm per decade. The values increased from about 180 mm in the 1950s to 270 mm in the last decade. As this analysis only considers annual values, no conclusion can be drawn regarding the occurrence of heavy rainfall events. In contrast to that, there is substantial evidence that the amount of snowfall is staying steady with only a very slight decrease of -2.7 mm per decade with a nearly constant climatology around 280 mm per year. The strongly increasing trend in rainfall and a negligible decreasing trend in snowfall indicate an overall warmer and wetter climate.

The incoming longwave radiation shows a decisive increasing trend in the last 70 years with 1.6 $\mathrm{W\,m^{-2}}$ per decade. The mean annual values increase on average from 227 $\mathrm{W\,m^{-2}}$ in the 1950s to 237 $\mathrm{W\,m^{-2}}$ in recent years. The increase in longwave radiation has a pronounced effect on the surface energy balance, with more energy being transferred to the ground surface.

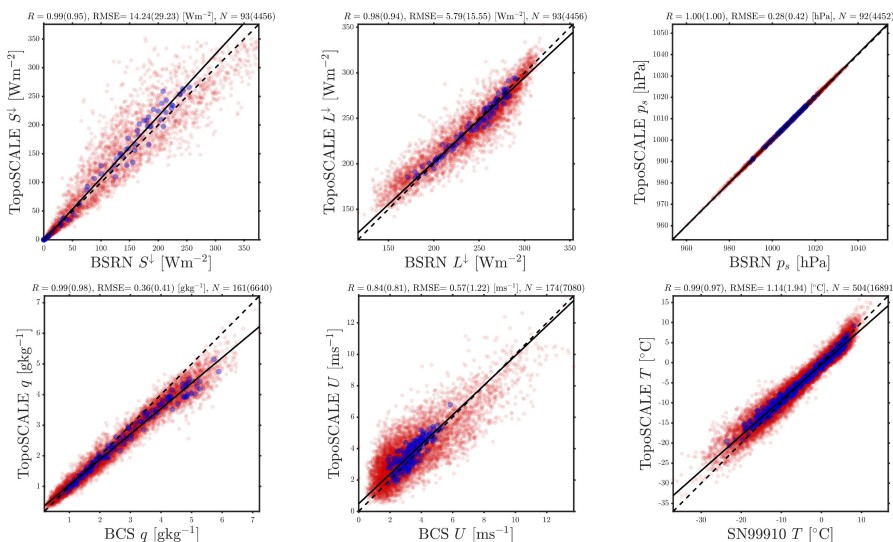

**Figure A2.** Validation of the downscaled data at stations (BSRN, BCS, SN99910) near Ny-Ålesund: Incoming shortwave (top left), incoming longwave (top middle), surface pressure (top right), specific humidity (bottom left), wind speed (bottom middle), and air temperature (bottom right). Red dots: daily averages; blue dots: monthly averages; dashed line: identity; solid line: linear best fit. Stations with the longest records were selected in each case. The correlation coefficient ($R$), RMSE, and number of points at the monthly (daily) scale are in the panel titles.

The steep slope angles of the coastal cliffs influence the incoming shortwave radiation. Mean annual values at Kjærsvika (representative field site for the southwest facing coastline) decrease strongly from 83 $\mathrm{W\,m^{-2}}$ to 78 $\mathrm{W\,m^{-2}}$ with -0.8 $\mathrm{W\,m^{-2}}$ per decade. With a substantial trend of -0.6 $\mathrm{W\,m^{-2}}$ per decade, Stuphallet shows a trend in the same order of magnitude, however, the incoming shortwave radiation is considerably lower, decreasing from 70 $\mathrm{W\,m^{-2}}$ to 67 $\mathrm{W\,m^{-2}}$. The reduced incoming shortwave radiation at Stuphallet is a result of the northeast exposition of the coastline resulting in longer periods of 570    shading.

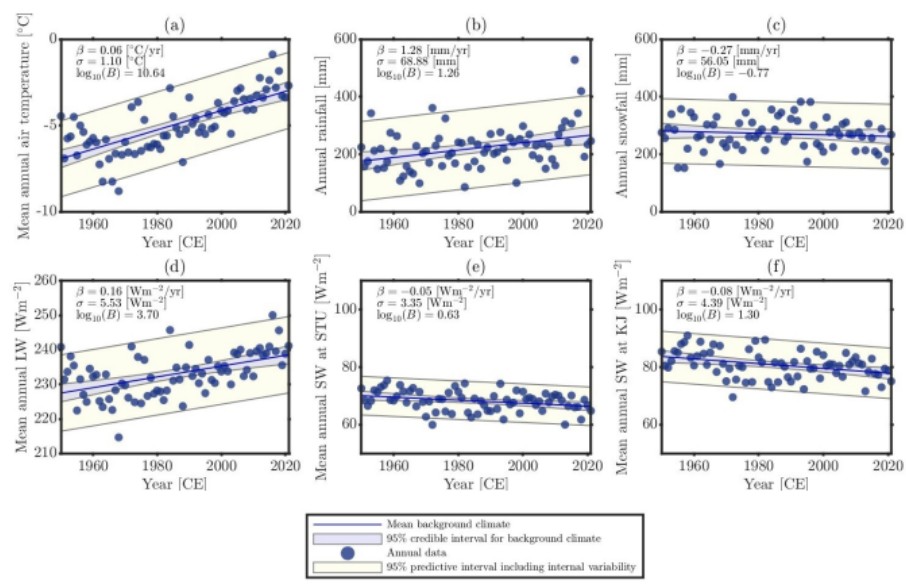

**Figure B1.** Climate on the Brøgger Peninsula for the last decades from downscaled from ERA5 data (a-f) and Ny-Ålesund station data (g): (a) Air temperature, (b) Rainfall, (c) Snowfall, (d) Incoming longwave radiation, (e) Incoming shortwave radiation at Stuphallet and (f) at Kjærsvika.

*Data availability.* The dGNSS survey and the analysis of the multiple digitization of the coastline are archived on zenodo (https://doi.org/ 10.5281/zenodo.7756973, Aga, 2023). The orthoimage of 2010 can be accessed at https://geodata.npolar.no/, whereas the orthoimages of 1970, 1990 and 2021 are not publicly available.

*Author contributions.* JA designed the concept of the study, conducted field work, performed the DSAS analysis and prepared the manuscript
with all tables and figures. LP contributed to the concept during all phases of the study and digitized the coastlines. SW provided ideas regarding the concept of the study as well as organizational and technical support. LP and AK acquired the aerial images. LG processed the aerial images. TE performed the post-processing of the dGNSS data. JA and KA performed the statistical tests. KA downscaled the climate data and prepared most of the appendix. All authors contributed to the final manuscript with input and suggestions.

*Competing interests.* There are no competing interests.

*Acknowledgements.* We acknowledge funding by EU Horizon 2020 ("Nunataryuk", grant no. 773421), the Research Council of Norway (Arctic Field Grant "Coastal erosion on the Brøgger peninsula, Svalbard", grant no. 321957 and "PCCH-Arctic", grant no. 320769). Furthermore, we acknowledge the support of Robin B. Zweigel during fieldwork.

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
