# Peer review of "Coastal retreat rates of high-Arctic rock cliffs on Brøgger peninsula, Svalbard, accelerate during the past decade"

_EGUsphere, 2023_

## Author Comment (AC1)

**Response to referee Gregor Luetzenburg**

We would like to thank the reviewer for evaluating our manuscript and for the useful comments, which helped to improve it. In the following, we provide a reply to the points discussed by the reviewer as well as changes in the manuscript.

The comments of the reviewer are written in **bold**, the extracts of the manuscript in *italics* with changes highlighted in blue and line numbers referring to the revised manuscript.

**The novelty of this work lies in measuring coastal cliff top retreat on Svalbard over 50 years on a high spatial resolution. Overall, most interpretations and conclusions are supported by the evidence, the assumptions are valid, the methodology is sound, the evidence is adequate, and the conclusions logically follow the results. However, the authors do not present any data to build a relationship between the measured retreat rates and potential environmental driving factors that can explain the increase in cliff top retreat and they cannot fulfill their third objective. Therefore, the final conclusion of the manuscript (retreat rates increase with climate change) remains highly speculative (although likely) and the manuscript lacks important scientific progress. Currently, the manuscript requires major improvements to make a substantial progress for the research field.**

**Overall, I agree with the comments from RC1. Therefore, I will focus on some suggestions to improve the manuscript. Although it takes an effort to incorporate all the suggested changes, I highly encourage the authors to go the extra mile and improve this manuscript. I think, if revised properly, this could become an important contribution for the field of Arctic coastal dynamics. I have six general suggestions to improve this study:**

**First, the authors mention coastline retreat throughout the manuscript, but they measure cliff top retreat. I agree with the authors that the cliff top can be detected more easily from historical aerial images, but I don't agree with the statement: 'we are confident that the shift of the top of the cliff is representative of the coastal retreat'. The authors do not provide a reference or any data to corroborate this statement. In contrast, a recent study has found that the cliff top is eroding with a higher magnitude, but a lower frequency compared to the cliff toe that is eroding with a lower magnitude but a higher frequency (Swirad et al. 2022, https://doi.org/10.1016/j.geomorph.2022.108318). Additionally, if the cliff is not a plunging cliff, which remains unclear in the authors description of the study site, the beach fronting the cliff plays an important role in the coastline retreat as well. The easiest solution would be to write cliff top retreat instead of coastline retreat throughout the manuscript. However, the most interesting solution would be to establish a relationship between coastline retreat and cliff toe as well as cliff top retreat. The latter is directly leading to the next point.**

We agree with the reviewer that a better explanation was needed why we used the cliff top retreat as a proxy for coastal erosion along Brøgger peninsula. To follow the suggestion of the reviewer, we added a new analysis in the revised manuscript. It is explained in the methods (Sect. 3.3), has an own new section in the results with two new figures (Sect. 4.2) and is mentioned at the beginning of the discussion (Sect. 5.1). The changes are as following:

Explanation in the methods:

Line 178: *The coastline was digitized along the top of the cliff (Fig. 1), which is slightly retreated compared to the actual shoreline, i.e. the boundary between water and land, due to unconsolidated sediments on top of the bedrock (Sect. 2). The top of the cliff has been used as a proxy for the shoreline in previous studies (e.g. Irrgang et al., 2018). However, it is important to note that the cliff top and the cliff foot can erode at different rates, and that the presence of frontal beaches can affect the erosion processes (Swirad and Young, 2022). To address this, we conducted an analysis to confirm the suitability of the cliff top retreat as a proxy for coastal retreat at our field site. To do so, we compared the distance between the cliff top and the shoreline, as well as the width of the frontal beaches for 53 cross-sections along the coast with a proximate distance of 100 m. Hereby, we used the orthoimages from 2010 and 2021 since the shoreline could not be reliably detected in the orthoimages from 1990 and 1970.*

The new section in the results:

Line 282: **4.2 Cliff top retreat as a proxy for coastal retreat**

*The analysis of 53 cross-sections along the investigated coastline of Brøgger peninsula shows that the distance between the cliff top and the shoreline changed only slightly by less than 0.10 m between 2010 and 2021. An example of a representative cross-section is given in Fig. 4. Here, the distance from the cliff top to the shoreline was 32.35 m in 2010 and reduced marginally to 32.19 m in 2021. Meanwhile, the width of the beach increased slightly from 2.82 m in 2010 to 3.05 m in 2021. This example showcases that the cliff top retreats at similar rates as the shoreline.*

*The mean distance for all cross-sections was 17.44 m in 2010 and reduced marginally to 17.35 m in 2021. The change in distance of 0.09 m is considerably smaller than the uncertainty associated with the digitization of the coastline (Table 2), i.e. the position of the cliff top. Also, the distribution of the distances is comparable between those two years with clusters of cliff top-shoreline-distances around 4 m, 12 m and 24 m (Fig. 5).*

*In addition, we analyzed the width of the frontal beaches along the 53 cross-sections. The results show that the width was only slightly reduced from 2.23 m in 2010 to 2.09 m in 2021. Furthermore, the characteristics of the cliff morphology did not change significantly: Seawater reaching the cliff foot directly was detected for 40 % (2010) and 42 % (2021) of the cross-sections, while 47 % (2010) and 45 % (2021) had frontal beaches with a high potential of inundation during stormy conditions (as the example given in Fig. 4), and only 7 % (2010, 2021) had extended frontal beaches, which could limit the effect of wave*

*activity on the cliff foot. Both the small change in beach width and cliff morphology are indicators, that the eroded material is*

*transported away effectively from the foot of the cliff and that no significant accumulation of eroded material occurs. Further-*

*more, we can conclude that the potential for wave activity is not affected due to changes in the cliff morphology over the years.*

*As our results suggest that the average distance between the cliff top and the shoreline or the width of the beach does not*

*change significantly over time, we are confident that the retreat of the cliff top is an applicable proxy for the coastal retreat at*

*our field site.*

[Figure]

*Figure 4. Cross-section with cliff top, cliff foot and shoreline for (a) 2010 and (b) 2021. The background shows the respective orthoimages. Source of the orthoimage: (a) Norwegian Polar Institute, https://geodata.npolar.no/; (b) Svalbard Integrated Arctic Earth Observing Systems SIOS, not publicly available.*

[Figure]

*Figure 5. Number of cross-sections that show a certain distance between cliff top and shoreline for 2010 and 2021. The distribution only changes slightly with clusters around 4* m*, 12* m *and 24* m*.*

At the beginning of the discussion, we added the following statement:

Line 370: *In this study, we analyzed the retreat of the cliff top along the lithified coast of Brøgger peninsula. We showed that this rate is representative of the coastal retreat (Sect. 4.2) so that a comparison with the erosion along other coastlines on Svalbard and in the Arctic can be drawn.*

**Second, the authors investigate cliff top retreat in 2D. Compared to low lying coastal environments like deltas and spits, coastal cliffs are unique because of their abrupt change in elevation. To fully understand the dynamics of coastal cliffs, including cliff toe, cliff face and cliff top retreat, the investigation requires 3D data. Based on cross-cliff elevation profiles derived from 3D elevation data, a relationship between cliff morphometrics (e.g. beach width and slope, cliff top and toe elevation, and cliff face slope) and cliff retreat rates can be built. Depending on those relationships, environ-**

**mental driving factors (overland flow, ground temperature, wave run-up, sea-ice content, sea-level change etc.) can be detected and changes over time can be discussed. 3D data (point clouds, DEMs etc.) of coastal cliffs in the Arctic is rare, but necessary to advance the understanding of coastal cliff dynamics in the Arctic. Without 3D change observations, this study lacks progress in the field as 2D retreat rates of cliffs are investigated already at other places on Svalbard which is discussed by the authors in chapter 5.1.**

We understand the point of the reviewer that it would be beneficial to include 3D data to differentiate between the cliff toe, cliff face and cliff top. However, such 3D data (DEMs or point clouds retrieved from photogrammetry or laser scanning) is not available for the field site and investigated time span.

As no 3D data was used in our study, the reviewer questions the progress of our work. We want to emphasize that the novelty of this study lies in the long-term retreat rates over 51 years, a time frame that has not been investigated in Svalbard before. Especially as coastal cliff dynamics can show a high variability throughout the years, a long time span contributes to the understanding of the coastal cliff top retreat on Svalbard. In contrast, previous studies have investigated coastal cliff erosion through terrestrial photogrammetry or laser scanning and by doing so, they retrieved 3D data, but could only cover short time spans
(we refer here to section 5.1 of our manuscript).

**Third, the current discussion chapter 5.2 presents no new insights because the authors present no data to discuss the potential environmental driving factors. In the current version of the chapter, the authors describe the conceptual model explaining how those cliffs erode which is already understood. What the scientific community is lacking is data showing**
**if the theoretical understanding is accurate and applicable. For example, the authors discuss potential differences in exposure to waves and storms of the northeast and southwest facing coastlines. The topographically downscaled ERA5 climate data could be used to create a wave rose of significant wave height. This would base the assumption in the manuscript on actual data and increase the overall strength of the manuscript, providing that the relationship between cliff retreat rate and environmental driving factors are understood. The same goes for other potential environmental**
**driving factors, e.g. provide actual data on sea-ice changes in the study area. The available data from the downscaled ERA5 data is very interesting, but the authors do not present the data in the main results. The discussion of these results is very short.**

We followed the suggestion of the reviewer and added a more detailed analysis of the potential environmental driving factors.
To do so, we added a new section in the methods (Sect. 3.5) and the results (Sect. 4.5) and extended significantly the discussion (Sect. 5.2). As the reviewer stated correctly, the exposure to waves and storms as well as the sea ice are the most important drivers due to their influence on mechanical abrasion. To base our assumptions on actual data as suggested by the reviewer, we added in the main manuscript data on the increase in storminess (here we took data from a station in ca. 8 km distance as the downscaled ERA5 data showed a large scatter) and an analysis of the increase in the potential fetch. All these factors are now
thoroughly discussed in Sect. 5.2.

Other minor influencing factors are presented in the appendix to not distract from the important discussion points in the main manuscript.

Changes in the methods:

Line 249: *3.5 Analysis of climate conditions*

[revised manuscript text omitted]

**Fourth, the only reason to include the rock surface temperature data in the manuscript is to directly couple it to cliff**
**retreat rates. It would be super interesting to show a relationship between seasonal cliff retreat and rock surface temperature (e.g. cliff retreat is faster when rock surface temperature is higher). Without this data, I would delete chapter 4.3.**

As we are investigating long time spans based on aerial images from 1970, 1990, 2010 and 2021, the temporal resolution
of our data does not allow us to analyze seasonal changes in cliff retreat. However, we are still convinced that the data on the rock surface temperature gives a better understanding of the local permafrost conditions. There are plenty of studies describing the permafrost conditions around Ny-Ålesund (ca. 8 km distance to our fieldsite), however, they are typically not considering the permafrost conditions in the rock walls. Schmidt et al. (2021) (https://doi.org/10.5194/tc-15-2491-2021) described rock surface temperatures close to Ny-Ålesund and showed that coastal cliffs can feature higher rock surface temperatures than in- land permafrost. The data presented in this study shows that the rock surface temperatures are falling into a temperature range with decreased stability (as discussed in Sect. 5.2). Therefore, adding this data on the permafrost conditions along Brøgger peninsula can be valuable for the interested reader, even though it is out of the scope of this study to link it to seasonal changes in cliff retreat.

**Fifth, in chapter 5.2 the authors discuss 'Coastal retreat rates under a warming climate'. Please rephrase to 'Coastal cliff retreat rates...'. It is generally understood that the warming climate is leading to the melting of the icecaps and glaciers which in turn will lead to rising sea-levels. However, Svalbard is experiencing postglacial isostatic rebound which might outpace the regional sea-level rise. More importantly, the melting of the Greenland Ice Sheet is leading to a gravitational sea-level lowering around Svalbard because of their proximity (< 2.200 km). Lastly, due to the melting**

**of the Greenland Ice Sheet, the rotational axis of the earth will move towards the area of mass loss which will also lead to a regional decrease of sea-levels around Svalbard. For more information see Slangen et al. 2022, e.g. figure 1 (https://doi.org/10.1175/JCLI-D-17-0110.1). Please include these considerations in your discussion on coastal cliff retreat rates under a warming climate as they are interconnected. You are welcome to get some inspiration from my recent paper discussing the same dynamic on a coastal cliff in Greenland (https://doi.org/10.1029/2022JF007026).**

    In the revised version, the showed that the cliff retreat rates are a suitable proxy for coastal retreat rates along Brøgger peninsula (Sect. 4.2) as suggested by the reviewer. Given the new information, we decided not to rephrase the title of this section.

    We included the isostatic rebound in the discussion of our revised manuscript and cited as well the two articles mentioned
by the reviewer.

    Line 448: *The increasing influence of wave activity due to intensified storminess and sea ice retreat may to some extent be counteracted by a lowering of relative sea level by -4.5±0.4 $\mathrm{mm\,a^{-1}}$ in Ny-Ålesund (Hanssen-Bauer et al., 2019). On Svalbard, the global mean sea level rise, caused mainly by changes in seawater density and land ice mass (Slangen et al.,*
*2017), is outpaced by the isostatic uplift of the land and changes in the gravitational field following deglaciation (Slangen et al., 2017; Kavan and Strzelecki, 2023). This phenomenon is dynamic in space and time (Hanssen-Bauer et al., 2019) and due to its heterogeneity, the effect on the coastal cliff erosion is difficult to quantify (Luetzenburg et al., 2023). However, we assume that the influence of the increasing wave activity on coastal cliff erosion is stronger than the relative sea level lowering.*

**Sixth, the manuscript lacks appropriate referencing of relevant literature and is presented as a regional case study with little interest for a broader, pan-Arctic audience. I would suggest putting a greater effort into discussing the broader applicability of the results for all of Svalbard and compare the findings with studies from similar Arctic coastlines (e.g. Greenland and Nunavut). Below some suggestions for literature to consider:**

The comparison and applicability of the study on Svalbard is discussed in Sect. 5.1. We followed the suggestion of the re-
viewer to widen the perspective and include more references from similar studies worldwide. To do so, we added information
in the introduction and placed our results in an international context in the discussion. The following changes were made in the
manuscript:

Line 21: *Therefore, Arctic coasts are often eroding more rapidly than coasts in temperate regions and the average retreat
rate is estimated to 0.5 $\mathrm{m\,a^{-1}}$ (Lantuit et al., 2012). However, the variability of coastal retreat rates across the Arctic is pro-
nounced, both on a regional and local scale. Lantuit et al. (2012) present a circum-Arctic database, where the highest rates
are detected in the Laptev Sea (0.73 $\mathrm{m\,a^{-1}}$), the East Siberian Sea (0.87 $\mathrm{m\,a^{-1}}$), the US Beaufort Sea (1.15 $\mathrm{m\,a^{-1}}$) and the
Canadian Beaufort Sea (1.12 $\mathrm{m\,a^{-1}}$). Numerous regional studies corroborate these numbers, for example with retreat rates
along the Bykovsky peninsula (Laptev Sea) of 0.59 $\mathrm{m\,a^{-1}}$ between 1951 and 2006 (Lantuit et al., 2011), along the US Beaufort
Sea of 1.8 $\mathrm{m\,a^{-1}}$ between 1940 and 2010 (Gibbs and Richmond, 2017) and along the Canadian Beaufort Sea with 0.7 $\mathrm{m\,a^{-1}}$
(Irrgang et al., 2018). The highest erosion rates are often found in ice-rich permafrost bluffs and barrier islands. Jones et al.
(2018) present a maximum of 48.8 $\mathrm{m\,a^{-1}}$ in such a setting along the US Beaufort Sea from 2007 to 2008.*

Line 376: *The calculated retreat rates are lower than the average change in Arctic coastlines of 0.5 $\mathrm{m\,a^{-1}}$ (Lantuit et al.,
2012). This is expected as high retreat rates are typically found along unlithified coasts, which account for 65 % of the Arctic
coastline (Irrgang et al., 2022). In contrast, the coastline along Brøgger peninsula is formed by bedrock, being characterized
by a higher resistance against mechanical abrasion compared to unlithified coasts, and the unconsolidated sediments on top
are not exposed to wave action. However, we detected higher retreat rates compared to other lithified coasts in the Arctic, e.g.
the Canadian Archipelago with 0.01 $\mathrm{m\,a^{-1}}$ (Lantuit et al., 2012). This can be explained by the long open water season at the
western coast of Svalbard (Sect. 5.2), resulting in the high importance of mechanical abrasion by wave action (Sect. 5.2). Other
contributing factors might be the highly fractured bedrock, decreasing the resistance of the material towards erosion and the
permafrost conditions, which show a temperature range with decreased bedrock stability (Sect. 5.2).*

We included the majority of the suggested papers in our revised manuscript.

**Barnhart, K. R., Overeem, I., & Anderson, R. S. (2014). The effect of changing sea ice on the physical vulnerability
of Arctic coasts. The Cryosphere, 8(5), 1777–1799. https://doi.org/10.5194/tc-8-1777-2014**

We included this reference in the revised manuscript.

**Boisson, A., Allard, M., & Sarrazin, D. (2020). Permafrost aggradation along the emerging eastern coast of Hudson
Bay, Nunavik (northern Que´bec, Canada). Permafrost and Periglacial Processes, 31(1), 128–140. https://doi.org/10.1002/ppp.2033**

As permafrost on Svalbard is known to be degrading, in contrast to the permafrost conditions presented in the suggested paper, we decided not to include this reference in our manuscript.

**Bourriquen, M., Baltzer, A., Mercier, D., Fournier, J., Pérez, L., Haquin, S., Bernard, E., & Jensen, M. (2016). Coastal evolution and sedimentary mobility of Brøgger Peninsula, northwest Spitsbergen. Polar Biology, 39(10), 1689-1698. https://doi.org/10.1007/s00300-016-1930-1**

We included this reference in the revised manuscript.

**Bourriquen, M., Mercier, D., Baltzer, A., Fournier, J., Costa, S., & Roussel, E. (2018). Paraglacial coasts responses to glacier retreat and associated shifts in river floodplains over decadal timescales (1966-2016), Kongsfjorden, Svalbard. Land Degradation & Development, 29(11), 4173-4185. https://doi.org/10.1002/ldr.3149**

We included this reference in the revised manuscript.

**Casas-Prat, M., & Wang, X. L. (2020a). Projections of extreme ocean waves in the Arctic and potential implications for coastal inundation and erosion. Journal of Geophysical Research: Oceans, 125(8), e2019JC015745. https://doi.org/10.1029/2019J**

We included this reference in the revised manuscript.

**Casas-Prat, M., & Wang, X. L. (2020b). Sea ice retreat contributes to projected increases in extreme Arctic Ocean surface waves. Geophysical Research Letters, 47(15), e2020GL088100. https://doi.org/10.1029/2020GL088100**

We included this reference in the revised manuscript.

**Jones, B. M., Irrgang, A., Farquharson, L. M., Lantuit, H., Whalen, D., Ogorodov, S., et al. (2020). Coastal permafrost erosion. https://doi.org/10.25923/e47w-dw52**

We included this reference in the revised manuscript.

**Kavan, J., & Strzelecki, M. C. (2023). Glacier decay boosts the formation of new Arctic coastal environments—Perspectives from Svalbard. Land Degradation & Development. https://doi.org/10.1002/ldr.4695**

We included this reference in the revised manuscript.

McCrystall, M. R., Stroeve, J., Serreze, M., Forbes, B. C., & Screen, J. A. (2021). New climate models reveal faster and larger increases in Arctic precipitation than previously projected. Nature Communications, 12(1), 6765. https://doi.org/10.1038/s414 021-27031-y

We included this reference in the revised manuscript.

Nielsen, D. M., Pieper, P., Barkhordarian, A., Overduin, P., Ilyina, T., Brovkin, V., et al. (2022). Increase in Arctic coastal erosion and its sensitivity to warming in the twenty-first century. Nature Climate Change, 12(3), 263–270. https://doi.org/10.1038/s41558-022-01281-0

We included this reference in the revised manuscript.

Sessford, E. G., Baeverford, M. G., & Hormes, A. (2015). Terrestrial processes affecting unlithified coastal erosion disparities in central fjords of Svalbard. Polar Research, 34(1), 24122. https://doi.org/10.3402/polar.v34.24122

We included this reference in the revised manuscript.

St-Hilaire-Gravel, D., Bell, T. J., & Forbes, D. L. (2010). Raised gravel beaches as proxy indicators of past sea-ice and wave conditions, Lowther Island, Canadian Arctic archipelago. Arctic, 63(2), 213–226. https://doi.org/10.14430/arctic976

The paper discusses gravel-dominated coastlines, so that we decided not to include this reference in our revised manuscript.

St-Hilaire-Gravel, D., Forbes, D. L., & Bell, T. (2012). Multitemporal analysis of a gravel-dominated coastline in the central Canadian Arctic archipelago. Journal of Coastal Research, 280, 421–441. https://doi.org/10.2112/jcoastres-d-11-00020.1

The paper discusses gravel-dominated coastlines, so that we decided not to include this reference in our revised manuscript.

Zago´rski, P., Rodzik, J., Moskalik, M., Strzelecki, M. C., Lim, M., Błaszczyk, M., et al. (2015). Multidecadal (1960–2011) shoreline changes in Isbjørnhamna (Hornsund, Svalbard). Polish Polar Research, 36(4), 369–390. https://doi.org/10.1515/popore-370    2015-0019

We included this reference in the revised manuscript.

**L9 Mention the cliff top retreat increase in percent between the three periods of investigation instead of the absolute values in the abstract.**

We think that the absolute values of coastal cliff retreat are important in the abstract, so that the reader knows the order of magnitude we are talking about. However, we see the point of the reviewer that the increase in percentage is also an important information. Therefore, we added this in the revised manuscript.

Line 8: *This is true for both the northeast facing coastline, with retreat rates increasing from $0.04\pm0.06\ \mathrm{m\,a^{-1}}$ (1970-1990) and $0.04\pm0.04\ \mathrm{m\,a^{-1}}$ (1990-2010) to $0.06\pm0.08\ \mathrm{m\,a^{-1}}$ (2010-2021) and the southwest facing coastline, where retreat rates of $0.26\pm0.06\ \mathrm{m\,a^{-1}}$ (1970-1990), $0.24\pm0.04\ \mathrm{m\,a^{-1}}$ (1990-2010) and $0.30\pm0.08\ \mathrm{m\,a^{-1}}$ (2010-2021) are measured. This corresponds to an increase in the most recent decade of 75 % for the northeast facing coastline and of 25 % for the southwest facing coastline.*

**L51 You would also detect long-term trends in coastal retreat if you would analyze the two areas together. The second half of the sentence 'by performing separate analyses for the northeast and southwest facing coastline' does not fit to the first half 'to detect long-term trends in coastal retreat along the Brøgger peninsula'.**

We changed the wording in this paragraph.

Line 70: *The main objectives of this study are (i) to detect long-term trends in coastal retreat along the Brøgger peninsula, separately analyzed for the northeast and southwest facing coastline, (ii) to analyze rock surface temperatures for both expositions of the coastline, and (iii) to link these changes to available climate data.*

**L55 It is essential for the readers understanding to mention that the study area is located in an uplifted beach ridge system and that the topmost layer of the cliffs consists of uplifted marine terraces. Please describe the Holocene landscape evolution history of the study and mention the current day uplift rates that are measured in Ny-Ålesund.**

In the revised manuscript, we mention the uplifted beach ridges and the current day uplift rates.

Line 80: *They are typically covered with several meters of unconsolidated sediments, consisting of raised beach ridges (Etzelmüller and Sollid, 1991), which are dated to the Late Weichsel (about $13.5\ \mathrm{ka}$) below $45\ \mathrm{m\,a.s.l.}$ (Forman et al., 1987) and uplifted following the isostatic rebound of the land caused by the retreat of glaciers (Rotem et al., 2023). The current uplift rate in Ny-Ålesund is $8.0\pm0.3\ \mathrm{mm\,a^{-1}}$ (Hanssen-Bauer et al., 2019).*

**L72 Did you try to use the aerial images from 1936 for cliff top mapping? Geyman et al. 2022 (https://doi.org/10.1038/s41586-021-04314-4) provides high resolution orthophotos in the supplement.**

We are aware of the aerial images provided by Geyman et al. (2022). It would be great to include data from 1936, however, with a resolution of 5 m, the cliff top cannot be digitized with the accuracy needed for this study. Therefore we decided, not to use this data in our study.

**L210 please just mention the retreat rates in the text and not the colors you assigned to the groups in fig 4.**

We followed the suggestion of the reviewer and changed the wording in the revised manuscript.

Line 317: *This is visualized in Fig. 6: while small retreat rates ($< 0.05\,\mathrm{m\,a^{-1}}$, little to no erosion) dominate in 1970-1990, the number of transects with higher retreat rates ($< 0.20\,\mathrm{m\,a^{-1}}$) increases along the entire coastline in 2010-2021.*

**L222 The last sentence of the paragraph should be moved to the discussion.**

We deleted the sentence from the result section. The point was added in the discussion with a different wording, so that it fits into the section.

Line 468: *The coastal cliffs stabilize the unconsolidated material on top. Analysis of DEMs generated by the orthoimages shows that the top of the cliff retreats typically in conjunction with the bedrock and consequently, the erosion of the sediments is highly dependent on the retreat of the bedrock below. Typically, retreat rates for single transects are in the same order of magnitude as the mean retreat rates. However, high retreat rates can occur locally and within only one time span. These values might be caused by large blocks being released in single events, which in return affect greatly the erosion of the overlying sediments and thus the cliff top.*

**L249 In my opinion it doesn't make sense to compare your very local retreat rate to an average pan-Arctic retreat rate. I would delete the sentence.**

As reviewer 1 asked for a comparison to international studies, we kept the sentence in the revised manuscript. However, we added some discussion to improve the clarity.

Line 376: *The calculated retreat rates are lower than the average change in Arctic coastlines of $0.5\,\mathrm{m\,a^{-1}}$ (Lantuit et al., 2012). This is expected as high retreat rates are typically found along unlithified coasts, which account for 65 % of the Arctic coastline (Irrgang et al., 2022). In contrast, the coastline along Brøgger peninsula is formed by bedrock, being characterized*

*by a higher resistance against mechanical abrasion compared to unlithified coasts, and the unconsolidated sediments on top are not exposed to wave action. However, we detected higher retreat rates compared to other lithified coasts in the Arctic, e.g. the Canadian Archipelago with 0.01* $\mathrm{m\,a^{-1}}$ *(Lantuit et al., 2012). This can be explained by the long open water season at the western coast of Svalbard (Sect. 5.2), resulting in the high importance of mechanical abrasion by wave action (Sect. 5.2). Other contributing factors might be the highly fractured bedrock, decreasing the resistance of the material towards erosion and the permafrost conditions, which show a temperature range with decreased bedrock stability (Sect. 5.2).*

**L260 reference the studies you are referring to.**

We added the references.

Line 395: *Previous studies of coastal cliff erosion on Svalbard determined the retreat rate with either terrestrial photogrammetry or laser scanning (Wangensteen et al., 2007; Lim et al., 2020).*

**L264 true, but you are also only looking at one geographically small, geologically homogenous area, and not several study sites all across the archipelago. I would rather investigate a smaller area in 3D than a larger area in 2D.**

The aim of our study is to present long-term retreat rates for coastal cliffs on Svalbard. For these time spans, we only have 2D data available (see answer to major reviewer comment above). Our approach gives the best approximation for long-term retreat rates with the available data.

**L312 Did you not observe an icefoot in spring in front of the cliff and what effect does this have for the cliff erosion dynamics?**

In the revised manuscript, we mention the development of ice foot.

Line 96: *In winter, an icefoot or snowdrift can develop, protecting the shore platform and the lower parts of the coastal rock wall from denudational processes.*

**L315 name the figure or chapter in the appendix you are referring to.**

We added this information throughout the entire revised manuscript when referring to information in the appendix.

Line 462: *Topographically downscaled ERA5 climate data (Fiddes and Gruber, 2014; Hersbach et al., 2020; Renette et al., 2023) for Brøgger peninsula show that air temperature and longwave radiation increased over the past decades, while*

*shortwave radiation decreased slightly (Appendix B).*

**L358 where are those sites? Please include coordinates and a map.**

We added the coordinates and a figure. As Stuphallet, Kongsfjordneset and Kjærsvika are already visualized in Fig. 1, we referred here to Fig. 1 and provided in addition a map for the three climate stations.

Line 522: *The terrain parameters were computed using a 30* m *resolution DEM of Svalbard (Norwegian Polar Institute, 2014a), focusing on 6 sites: the Ny-Ålesund climate station SN99910 (78°55'23" N, 11°55'55" E), the Ny-Ålesund BSRN station (78°55'22" N, 11°55'38" E), the Bayelva soil and climate station BCS (78°55'15" N, 11°50'0" E) (Fig. A1), Stuphallet (78°57'57" N, 11°35'57" E), Kongsfjordneset (78°58'21" N, 11°30'45" E), and Kjærsvika (78°55'17" N, 11°25'11" E) (Fig. 1).*

[Figure]

***Figure A1.** (a) Orthoimage of the surroundings of Ny-Ålesund, where the climate stations are located, which were used for validation of the downscaled atmospheric data. Source: NP_Basiskart_Svalbard_WMTS_32633 © Norwegian Polar Institute.*

**Figure 1 please indicate the photo positions of panel b and c in panel a.**

We photos were both taken at Stuphallet. As this point is already marked, we added this information in the figure caption.

Caption of Fig. 01: *The (b) top and the (c) bluff face along the coastline of Brøgger peninsula show the rock cliffs covered with unconsolidated sediments. The green arrows and lines indicate the top of the cliff that is digitized.* *Both photos were taken*

*at the location of the dGNSS survey at Stuphallet.*

**Figure 4 the color scheme is not very intuitive. I would assume green is stable and yellow moderate retreat rates. For the highest rates of erosion, I would assume red colors.**

We agree with the reviewer that a color ramp from green to red would be most intuitive. However, due to the guidelines for colorblindness, we cannot follow the suggestion of the reviewer. Therefore, we changed the color ramp, so that green indicates stable conditions (not yellow as before) and blue shows the highest retreat rates.

[Figure]

| Coastal retreat rates [m/a] | | | |
|---|---|---|---|
| < 0.05 | 0.15 - 0.20 | 0.30 - 0.35 | 0.45 - 0.50 |
| 0.05 - 0.10 | 0.20 - 0.25 | 0.35 - 0.40 | > 0.50 |
| 0.10 - 0.15 | 0.25 - 0.30 | 0.40 - 0.45 | |

---

## Author Comment (AC2)

**Response to anonymous referee 1**

We would like to thank the reviewer for the detailed and useful comments, which have helped to improve the quality and readability of our manuscript. In the following, we provide a reply to the points discussed by the reviewer as well as changes in the manuscript.

The comments of the reviewer are written in **bold**, the extracts of the manuscript in *italics* with changes highlighted in blue and line numbers referring to the revised manuscript.

**The manuscript deals with a relevant and up-to-date topic concerning coastal retreat rates of Arctic rock cliffs located on the Brøgger peninsula in Svalbard, Norway. In times of ongoing and projected future climate change more detailed**
**scientific results on coastal erosion particularly in permafrost regions are indeed necessary. The main aim of the study is to detect long-term trends in differently exposed coastlines and linking these changes to available climate date. Results are achieved by using high-resolution orthoimagery combined with dGNSS measurements. In addition a short term dataset of rock surface temperatures was acquired and topography-based downscaling of atmospheric reanalysis data was utilized. While the presented results elucidate the contrasting trends of retreat rates on the northeast and southwest**
**facing coastlines very nicely, the given explanations for the detected differences as well for the potential main climatic drivers are too general and leave a lot of room for improvement. A one-year dataset of rock surface temperatures is certainly quite short for a more comprehensive temperature analysis.**

**The manuscript is well presented and has a logical structure. All tables and figures including the appendix are well**
**prepared. While the method and result sections are quite detailed, the introduction and discussion sections need to be clearly revised and improved. In its current state, the manuscript has the character of a regional case study with limited novelty for a broader and international readership. I recommend to widen the author's perspective from Svalbard also to other Arctic coastal areas worldwide and to include references of similar or comparable studies from these regions too. A short compilation of published coastal retreat rates in the Arctic would certainly be interesting. I have listed**
**some international references at the end that the authors might find useful. The study site description needs to be revised significantly. For readers unfamiliar with Svalbard, the given study site description is not helpful at all. Please see my detailed comments below. The discussion section needs to be significantly revised as well. At the moment, I can't see a wider geoscientific relevance of the presented research. In addition, by placing your results achieved in an international context, you will definitely reach a larger readership. Under 5.2 it might be worth to discuss actual and potential**
**implications of an increased susceptibility of the coastline. For instance what does enhanced coastal erosion mean for the population and infrastructure of Svalbard? I think the manuscript could benefit from illuminating these points in a more detailed and broader perspective as well as to discuss possible implications of the achieved findings in more detail**

**and in an international context.**

We followed the suggestions of the reviewer and revised the manuscript with a focus on the introduction and the discussion:

- Introduction: We widened the perspective and included more references from similar studies. In particular, we added a compilation of published coastal retreat rates from Arctic coastlines worldwide. Here, we included the references suggested by the reviewer. Please find the changes in the revised manuscript below.

- Study site description: We added the information suggested by the reviewer. For a detailed answer, please refer to the detailed comments below.

- Discussion: We placed our results in an international context as suggested by the reviewer. Please find the changes in the revised manuscript below.

- Discussion: We extended the explanations for the detected differences between the northeast and southwest facing coastline as suggested by the reviewer. Please find the changes in the revised manuscript below.

- Methods, results and discussion: We followed the suggestion of the reviewer and added a more detailed analysis of the potential main climatic drivers. To do so, we added a new section in the methods (Sect. 3.5) and the results (Sect. 4.5) and extended significantly the discussion (Sect. 5.2). Hereby, we focused on the wind regime and the potential fetch, as these have a main influence on the mechanical abrasion, which is expected to have the strongest influence on erosion along the coastline of Brøgger peninsula. Other minor factors are presented in the appendix to not distract from the important discussion points in the main manuscript.

- We agree with the reviewer that the one-year data set of rock surface temperatures is short. However, it shows that the rock surface temperatures are falling into a temperature range with decreased stability (as discussed in Sect. 5.2). Therefore, adding this data on the permafrost conditions along Brøgger peninsula can be valuable for the interested reader.

- We included a paragraph about the impacts on settlements and infrastructure as suggested by the reviewer. Please find the changes in the revised manuscript below.

Changes in the introduction to widen the perspective with a compilation of published coastal retreat rates from Arctic coastlines:

Line 21: *Therefore, Arctic coasts are often eroding more rapidly than coasts in temperate regions and the average retreat rate is estimated to $0.5 \, \mathrm{m \, a^{-1}}$ (Lantuit et al., 2012). However, the variability of coastal retreat rates across the Arctic is pronounced, both on a regional and local scale. Lantuit et al. (2012) present a circum-Arctic database, where the highest rates are detected in the Laptev Sea ($0.73 \, \mathrm{m \, a^{-1}}$), the East Siberian Sea ($0.87 \, \mathrm{m \, a^{-1}}$), the US Beaufort Sea ($1.15 \, \mathrm{m \, a^{-1}}$) and the*

*Canadian Beaufort Sea (1.12* $\mathrm{m\,a^{-1}}$*). Numerous regional studies corroborate these numbers, for example with retreat rates along the Bykovsky peninsula (Laptev Sea) of 0.59* $\mathrm{m\,a^{-1}}$ *between 1951 and 2006 (Lantuit et al., 2011), along the US Beaufort Sea of 1.8* $\mathrm{m\,a^{-1}}$ *between 1940 and 2010 (Gibbs and Richmond, 2017) and along the Canadian Beaufort Sea with 0.7* $\mathrm{m\,a^{-1}}$ *(Irrgang et al., 2018). The highest erosion rates are often found in ice-rich permafrost bluffs and barrier islands. Jones et al. (2018) present a maximum of 48.8* $\mathrm{m\,a^{-1}}$ *in such a setting along the US Beaufort Sea from 2007 to 2008.*

Changes in the discussion to place the results in an international context:

Line 376: *The calculated retreat rates are lower than the average change in Arctic coastlines of 0.5* $\mathrm{m\,a^{-1}}$ *(Lantuit et al., 2012). This is expected as high retreat rates are typically found along unlithified coasts, which account for 65 % of the Arctic coastline (Irrgang et al., 2022). In contrast, the coastline along Brøgger peninsula is formed by bedrock, being characterized by a higher resistance against mechanical abrasion compared to unlithified coasts, and the unconsolidated sediments on top are not exposed to wave action. However, we detected higher retreat rates compared to other lithified coasts in the Arctic, e.g. the Canadian Archipelago with 0.01* $\mathrm{m\,a^{-1}}$ *(Lantuit et al., 2012). This can be explained by the long open water season at the western coast of Svalbard (Sect. 5.2), resulting in the high importance of mechanical abrasion by wave action (Sect. 5.2). Other contributing factors might be the highly fractured bedrock, decreasing the resistance of the material towards erosion and the permafrost conditions, which show a temperature range with decreased bedrock stability (Sect. 5.2).*

Changes in the discussion for a more detailed explanation of the detected differences between the northeast and southwest facing coastline:

Line 429: *However, the impact of windiness and wind-induced wave action is expected to vary along the coastline of Brøgger peninsula. The northeast facing coastline is characterized by a relatively sheltered position within the Kongsfjorden system, with land in most cardinal directions within a range of 10 to 15* km*. This likely restricts the fetch and consequently wave activity, which may explain the lower coastal retreat rates in this sector. In contrast, the southwest facing coastline is more exposed to the open sea, especially in westerly and northwesterly directions in which the potential wave fetch is controlled by the distance to the sea ice edge in Fram Strait, which has clearly increased in the last decades (Sect. 4.5). Previous studies have shown that an increasing fetch results in wave growth (Casas-Prat and Wand, 2020b), increasing the capability for wave-driven erosion (Casas-Prat and Wang, 2020a). Therefore, the increasing distance to the sea ice edge towards the open Fram Strait likely increases the wave activity and thus mechanical abrasion along southwest sector of Brøgger peninsula, likely explaining the higher erosion rates found here.*

Changes in the revised manuscript for a more detailed analysis of the potential main climatic drivers:

Changes in the methods:

Line 249: **3.5 Analysis of climate conditions**

[revised manuscript text omitted]

*are likely, with mostly ice-free conditions in the last decade.*

The following information about the impacts on human infrastructure were added in the revised manuscript:

Line 484: *With no infrastructure along the investigated coastline of Brøgger peninsula, the consequences for the population*

*are expected to be minor. Widening the perspective on Svalbard, settlements and infrastructure are concentrated along lithified*

*and unlithified coastlines, such as the major administrative center Longyearbyen. Jaskolski et al. (2018) identified ports, quays and structures for coastal protection as the most endangered infrastructure along the coast in Longyearbyen, where not only coastal retreat but also river erosion, permafrost thaw and solifluction play an important role. In addition, the cultural heritage of Svalbard is severely affected by coastal erosion (Nicu et al., 2020, 2021). Pan-Arctic, increased coastal erosion threatens*

*Arctic communities, e.g. in Alaska (Bronen and Chapin III, 2013; Brady and Leichenko, 2020) and Canada (Andrachuk and Pearce, 2010; Radosavljevic et al., 2016; Irrgang et al., 2019), and even forced relocation of some settlements (Bronen and Chapin III, 2013). As the results from our study show, this may not only concern unlithified, ice-rich coasts, which are known to be highly affected by coastal erosion (Irrgang et al., 2022), but also lithified coasts in the Arctic.*

**Under point 2: Some more details on the climatic setting of your study site are needed here, e.g.:**

**- geographical coordinates of the study sites**

We added the coordinates of the field site.

Line 75: *The study area is the northwestern part of the Brøgger peninsula, located on the west coast of Spitsbergen. It stretches from the southwest at approximately 78°55' N, 11°15' E to the northeast at 78°59' N, 11°40' E.*

**- mean annual air temperature for the period of recorded data**

We added data on the mean annual air temperature.

Line 103: *Station data shows that air temperatures increased from an average of -5.9°C in the 1970s to -3.1°C in the 2010s (Norwegian Meteorological Institute, 2022b). This corresponds to a linear increase in mean annual air temperature for the*

*time period 1971 to 2017 of 0.71 °C per decade, with the strongest increase in the winter season of 1.35 °C per decade (Hanssen-Bauer et al., 2019). Maturilli et al. (2015), who looked at a shorter time period from 1994 to 2013, detected an even stronger trend of 1.3±0.7 °C per decade, with a winter warming of 3.1±2.6 °C per decade. Hereby, the month with the lowest air temperatures is typically February, while the highest values are found in July (Hanssen-Bauer et al., 2019). The observed winter warming is associated with an increase in incoming longwave radiation of 15.6±11.6 $\mathrm{W\,m^{-2}}$ per decade. As the du-*

*ration of snow cover is shortened and, hence, the reflection of shortwave radiation is reduced, the net shortwave radiation is slightly increased in the spring and summer seasons (Maturilli et al., 2015).*

**- any information about the wind regime**

We added information on the wind regime.

    Line 119: *The wind regime of Brøgger peninsula is notably influenced by the mountainous terrain, the topography of Kongs-fjorden and katabatic winds originating from the glaciers, resulting in a complex wind field (Svendsen et al., 2002; Maturilli and Kayser, 2017). In Ny-Ålesund, the occurrence of days, where mean hourly wind speeds with a strong breeze or stronger*
*(wind speeds >= 10.8 m s$^{-1}$) have been recorded, has increased from an average of around 65 days in the 1970s to 90 days in recent years (as described in Sect. 4.5; Norwegian Centre for Climate Services, 2023). It is important to note that the wind characteristics in Ny-Ålesund are not directly applicable to the study site because the influence of the mountains and the fjord diminishes and likely plays a lesser role. At the tip of Brøgger peninsula, which is part of the study site, wind speed measurements are available since the summer of 2021 (Norwegian Centre for Climate Services, 2023). These data reveal stronger wind*
*speeds, with 112 days with strong breezes recorded in 2022, compared to 84 days in Ny-Ålesund, but long-term trends in wind speeds for the investigated field site are not available.*

**- general information on snow cover and more specific information on the mentioned reduced snow cover**

We added more information on the snow cover.

    Line 112: *The mean annual precipitation between 2010 and 2021 in Ny-Ålesund was 526 mm, showing an increasing trend in the last decades since the 1980s with 384 mm (Norwegian Meteorological Institute, 2022a). Both snowfall and rainfall can occur at any time during the year, but the snow season typically lasts from October to June (Hop and Wiencke, 2019). Follow-*
*ing the trend of warming air temperatures, a shifting of the onset of snow melt to earlier dates is observed with -5.8±8.3 days per decade (Maturilli et al., 2015). At the Bayelva soil and climate station typical snow depths between 0.65 and 1.4 m are observed (Boike et al., 2018). The steep coastal cliffs of Brøgger peninsula are typically snow-free. Here, snow accumulations are limited to edges in the bedrock and snow accumulations at the foot of the rock walls (Schmidt et al., 2021).*

**- permafrost distribution**

    We added information on the permafrost characteristics.

    Line 87: *The field site is characterized by continuous permafrost, even though the presence of taliks cannot be excluded. At*
*the Bayelva soil and climate station, which is about 8 km distance to the investigated field site, mean annual ground temperatures in a depth of 9 m are recorded with -3.0 to -2.6°C between 2009 and 2016 (GTN-P, 2018). Measurements of rock surface temperatures in the coastal cliffs of Brøgger peninsula in about 8 km distance to the field site also revealed relatively warm permafrost, with annual values between -0.6 and -3.6°C in the years 2016 to 2020 (Schmidt et al., 2021).*

**- important denudational processes**

We added information on the denudational processes.

Line 92: *The bedrock cliffs on Svalbard are exposed to several denudational processes. At the shore platform and the lower part of the coastal cliff, abrasion is likely the main controlling factor with wave action acting upon the bedrock (Are, 1988a, b), redistributing beach sediments and consequently polishing the bedrock (Strzelecki et al., 2017). In addition, wetting-drying cycles by tidal water level changes and freeze-thaw processes can weaken the bedrock, especially where open cracks are present. In winter, an icefoot or snowdrift can develop, protecting the shore platform and the lower parts of the coastal rock wall from denudational processes. Above, where waves cannot reach the coastal cliff, periglacial weathering is controlling the erosion (Strzelecki et al., 2017). Here, rock fracturing through ice segregation may contribute to an increased susceptibility of the bedrock (Ødegård and Sollid, 1993). The unconsolidated sediments on top of the bedrock rest with their natural friction angle, following the erosion of the bedrock.*

**I think it is necessary to explain the permafrost situation of your study site in more detail here. For readers who are not familiar with the study site or the special conditions of Svalbard, it is not clear how the permafrost distribution looks like in Svalbard and where and how it is measured. What is the specific permafrost situation along the coastal rock cliffs?**

We added information about the permafrost conditions at our field site. Please see above.

**Concerning the actual cliff study sites:**

**- information of the entire actual cliff height (if possible differentiated in bedrock height and the height of the unconsolidated marine deposits on top)**

We added the information on the cliff height including the bedrock height.

Line 84: *The coastal cliffs have a mean height of 15.5 m with a maximum of 28.0 m, whereof the bedrock accounts for approximately 10.5 m on average.*

**- information on crack and fracture density if available**

Unfortunately, there are no data available for the crack and fracture density of the coastal cliffs of Brøgger peninsula. Own observations are restricted by the difficult access to the shoreline. The only statement that we could find is that they are "gen- erally highly fractured". We included this statement in the manuscript.

Line 78: *The bedrock is* *typically highly fractured (Ødegård and Sollid, 1993) and* *dominated by conglomerates, sandstones,*
*shales and carbonates from the Carboniferous to Permian age, which often form overhanging cliffs.*

**Under point 3: The "accuracy and error analysis" is clearly structured and reasonable explained. It is obvious that**
**the orthoimages from 1970 are the most critical ones. However, the shown example of the digitized coastline from 1970**
**is not clear to me. Why is the digitized line so far away from the "coastline" and how did you manage to recognize the**
**notches?**

Thank you for your comment. We agree that the quality of the 1970 orthoimage can significantly impact the accuracy of the
digitized coastline. We have updated the main text as follows.

Line 186: *The* *cliff top* *was digitized manually in a GIS environment using the WGS 1984 UTM Zone 33N at a scale of 1:400*
*by the same operator.* *The digitization process relied on visually interpreting the coastline from the orthophoto, with additional*
*visual support from topographic maps, including hillshaded DEM and slope.*

Line 190: *The* *digitization along the bedrock coast was interrupted sporadically due to (1) rivers feeding into the fjord,*
*which incised* *into the bedrock, (2) closely spaced thermo-erosional gullies, which* *prevented* *a clear detection of the coastline,*
*and (3) the quality of the orthoimages. The last point* *affected* *only the digitization of the southwest facing coastline in the 1970*
*data,* *a challenging area due to* *unfavourable illumination conditions* *and excessive blurring of the orthoimage. In areas where*
*the 1970 orthoimage met acceptable quality standards (i.e., exhibited less blur and higher contrast), we mapped the coastline*
*and the notches by tracing the boundary between the dark and light grey areas (as shown in Fig. 3). In this case, we assumed*
*that the lighter area in the orthoimage corresponded to the steeper terrain of the cliffs. The hillshade and slope maps were very*
*noisy and thus were not considered in the digitization process.*

**Figure 4: This is a very nice figure but it is a bit confusing that the retreat rates for all three time intervals are shown**
**on the same orthoimage. I can understand the decision in order to have a good visibility. However, you could maybe**
**mention the actual date of the orthoimage for clarification.**

As the reviewer mentions, we tried in the beginning to use different orthoimages in the background, but that disturbed the
visibility of the results. We added the date of the orthoimage in the caption.

Caption of Figure 6: *Cliff top* *retreat rates at the northeast and southwest facing coastline of Brøgger peninsula for the time*
*periods 1970-1990, 1990-2010 and 2010-2021.* *The background shows the orthoimage of 2021.* *Source of the orthoimage:*

*Svalbard Integrated Arctic Earth Observing Systems SIOS, not publicly available.*

**Under point 4.2: It would certainly be interesting if you could elaborate on possible reasons for the different detected retreat rates on the NE and SW facing side in more detail, as this is one of your main findings.**

We extended our discussion about the detected differences along the northeast and southwest facing coastline. However, this was not done in Sect. 4.2 as this section is describing the results, but in the discussion under Sect. 5.2. The manuscript was changed as following:

Line 429: *However, the impact of windiness and wind-induced wave action is expected to vary along the coastline of Brøgger*
*peninsula. The northeast facing coastline is characterized by a relatively sheltered position within the Kongsfjorden system, with land in most cardinal directions within a range of 10 to 15* km. *This likely restricts the fetch and consequently wave activity, which may explain the lower coastal retreat rates in this sector. In contrast, the southwest facing coastline is more exposed to the open sea, especially in westerly and northwesterly directions in which the potential wave fetch is controlled by the distance to the sea ice edge in Fram Strait, which has clearly increased in the last decades (Sect. 4.5). Previous stud-*
*ies have shown that an increasing fetch results in wave growth (Casas-Prat and Wand, 2020b), increasing the capability for wave-driven erosion (Casas-Prat and Wang, 2020a). Therefore, the increasing distance to the sea ice edge towards the open Fram Strait likely increase the wave activity and thus mechanical abrasion along southwest sector of Brøgger peninsula, likely explaining the higher erosion rates found here.*

**Under point 4.3: Please see comment under point 2.**

We agree with the reviewer that the one-year data set of rock surface temperatures is short. However, it shows that the rock surface temperatures are falling into a temperature range with decreased stability (as discussed in Sect. 5.2). Therefore, adding this data on the permafrost conditions along Brøgger peninsula can be valuable for the interested reader.

**Line 230: from 1 September 2020 to 31 August 2021**

We corrected the spelling error in the manuscript.

Line 335: *The measurement period of the logger RW-SW close to Kjærsvika lasted from 1 September 2020 to 31 August  2021.*

**Line 342: September 2020 to August 2021?**

Of course! We corrected it in the manuscript.

     Line 506: *Installed temperature loggers recorded rock surface temperatures in the time period September 2020 to August 2021.*

**Fig. 4: "however" is redundant here**

     We removed "however" in the figure caption.

     Fig. 6: *Source of the orthoimage: Svalbard Integrated Arctic Earth Observing Systems SIOS,  not publicly available.*

     We decided to not include this reference in our manuscript, as it describes a storm surge inundating low-lying tundra plains, while the coastal cliffs in this study have an average height of 15.5 m, so that similar events are not expected here.

---

## Author Comment (AC3)

**Response to anonymous referee 2**

We would like to thank the reviewer for evaluating our manuscript and for the useful comments, which helped to improve it. In the following, we provide a reply to the points discussed by the reviewer as well as changes in the manuscript.

The comments of the reviewer are written in **bold**, the extracts of the manuscript in *italics* with changes highlighted in blue and line numbers referring to the revised manuscript.

The manuscript focuses on cliff erosion along 5.5 km coastline in NW Svalbard. Four aerial surveys were used to derive decadal-scale cliff retreat rates. The study is valuable given limited research on Arctic rock coasts, in particular at the timescales exceeding a few years.

10

25

30

I think that the paper is well-written with very clear methods. I really appreciate error estimation. The results are concrete. The figures are of good quality. There is no redundant information.

I agree with some conserns by Referee 1 and will not re-list them here. Definitely the reference to beyond-Svalbard Acrtic cliff studies is missing, so is a more detailed study area description including cliff morphology (cliff height, typical slope of bedrock wall and overlaying soft sediment). I would expect better justification of using top of the overlaying material as the proxy for coastline, given extensive discussion on this topic in rock coast literature. In terms of limited analysis on environmental conditions influencing coastal erosion raised by Refree 1, I would say you can go either way - perform more in-depth analyses or consider it out of scope of the study (but then I would move the temperature data to appendix and take the interpretation out of conclusions). Indeed, as of now there is quite a mismatch between mea-

20 to appendix and take the interpretation out of conclusions). Indeed, as of now there is quite a mismatch between measuring erosion and environmental conditions.

We widened the perspective and included more references from similar studies worldwide. To do so, we added information in the introduction and placed our results in an international context in the discussion. The following changes were made in the manuscript:

Line 21: Therefore, Arctic coasts are often eroding more rapidly than coasts in temperate regions and the average retreat rate is estimated to 0.5 m a-1 (Lantuit et al., 2012). However, the variability of coastal retreat rates across the Arctic is pronounced, both on a regional and local scale. Lantuit et al. (2012) present a circum-Arctic database, where the highest rates are detected in the Laptev Sea (0.73 m a-1), the East Siberian Sea (0.87 m a-1), the US Beaufort Sea (1.15 m a-1) and the Canadian Beaufort Sea (1.12 m a-1). Numerous regional studies corroborate these numbers, for example with retreat rates along the Bykovsky peninsula (Laptev Sea) of 0.59 m a-1 between 1951 and 2006 (Lantuit et al., 2011), along the US Beaufort Sea of 1.8 m a-1 between 1940 and 2010 (Gibbs and Richmond, 2017) and along the Canadian Beaufort Sea with 0.7 m a-1

(Irrgang et al., 2018). The highest erosion rates are often found in ice-rich permafrost bluffs and barrier islands. Jones et al. (2018) present a maximum of 48.8 m  $a^{-1}$  in such a setting along the US Beaufort Sea from 2007 to 2008.

Line 376: The calculated retreat rates are lower than the average change in Arctic coastlines of 0.5 m a-1 (Lantuit et al., 2012). This is expected as high retreat rates are typically found along unlithified coasts, which account for 65 % of the Arctic coastline (Irrgang et al., 2022). In contrast, the coastline along Brøgger peninsula is formed by bedrock, being characterized

- 40 by a higher resistance against mechanical abrasion compared to unlithified coasts, and the unconsolidated sediments on top are not exposed to wave action. However, we detected higher retreat rates compared to other lithified coasts in the Arctic, e.g. the Canadian Archipelago with  $0.01 \text{ m a}^{-1}$  (Lantuit et al., 2012). This can be explained by the long open water season at the western coast of Svalbard (Sect. 5.2), resulting in the high importance of mechanical abrasion by wave action (Sect. 5.2). Other contributing factors might be the highly fractured bedrock, decreasing the resistance of the material towards erosion and the
- 45 *permafrost conditions, which show a temperature range with decreased bedrock stability (Sect. 5.2).*

We extended the description of the study site (please refer to the track-changes-document for an entire compilation of the changes), including additional information about the cliff morphology.

50 Line 84: The coastal cliffs have a mean height of 15.5 m with a maximum of 28.0 m, whereof the bedrock accounts for approximately 10.5 m on average. The average slope angle of the unconsolidated sediments is approximately 35°.

We agree with the reviewer that a better explanation was needed why we used the cliff top retreat as a proxy for coastal erosion along Brøgger peninsula. To follow the suggestion of the reviewer, we added a new analysis in the revised manuscript. It
is explained in the methods (Sect. 3.3), has an own new section in the results with two new figures (Sect. 4.2) and is mentioned at the beginning of the discussion (Sect. 5.1). The changes are as following:

Explanation in the methods:

35

60 Line 178: The coastline was digitized along the top of the cliff (Fig. 1), which is slightly retreated compared to the actual shoreline, i.e. the boundary between water and land, due to unconsolidated sediments on top of the bedrock (Sect. 2). The top of the cliff has been used as a proxy for the shoreline in previous studies (e.g. Irrgang et al., 2018). However, it is important to note that the cliff top and the cliff foot can erode at different rates, and that the presence of frontal beaches can affect the erosion processes (Swirad and Young, 2022). To address this, we conducted an analysis to confirm the suitability of the cliff

65 top retreat as a proxy for coastal retreat at our field site. To do so, we compared the distance between the cliff top and the shoreline, as well as the width of the frontal beaches for 53 cross-sections along the coast with a proximate distance of 100 m. Hereby, we used the orthoimages from 2010 and 2021 since the shoreline could not be reliably detected in the orthoimages

**from 1990 and 1970.**

70 The new section in the results:

**Line 282: 4.2 Cliff top retreat as a proxy for coastal retreat**

The analysis of 53 cross-sections along the investigated coastline of Brøgger peninsula shows that the distance between the 75 cliff top and the shoreline changed only slightly by less than 0.10 m between 2010 and 2021. An example of a representative cross-section is given in Fig. 4. Here, the distance from the cliff top to the shoreline was 32.35 m in 2010 and reduced marginally to 32.19 m in 2021. Meanwhile, the width of the beach increased slightly from 2.82 m in 2010 to 3.05 m in 2021. This example showcases that the cliff top retreats at similar rates as the shoreline.

- 80 The mean distance for all cross-sections was 17.44 m in 2010 and reduced marginally to 17.35 m in 2021. The change in distance of 0.09 m is considerably smaller than the uncertainty associated with the digitization of the coastline (Table 2), i.e. the position of the cliff top. Also, the distribution of the distances is comparable between those two years with clusters of cliff top-shoreline-distances around 4 m, 12 m and 24 m (Fig. 5).
- In addition, we analyzed the width of the frontal beaches along the 53 cross-sections. The results show that the width was only slightly reduced from 2.23 m in 2010 to 2.09 m in 2021. Furthermore, the characteristics of the cliff morphology did not change significantly: Seawater reaching the cliff foot directly was detected for 40 % (2010) and 42 % (2021) of the cross-sections, while 47 % (2010) and 45 % (2021) had frontal beaches with a high potential of inundation during stormy conditions (as the example given in Fig. 4), and only 7 % (2010, 2021) had extended frontal beaches, which could limit the effect of wave activity on the cliff foot. Both the small change in beach width and cliff morphology are indicators, that the eroded material is
- transported away effectively from the foot of the cliff and that no significant accumulation of eroded material occurs. Furthermore, we can conclude that the potential for wave activity is not affected due to changes in the cliff morphology over the years.

As our results suggest that the average distance between the cliff top and the shoreline or the width of the beach does not 95 change significantly over time, we are confident that the retreat of the cliff top is an applicable proxy for the coastal retreat at our field site.

Figure 4. Cross-section with cliff top, cliff foot and shoreline for (a) 2010 and (b) 2021. The background shows the respective orthoimages. Source of the orthoimage: (a) Norwegian Polar Institute, https://geodata.npolar.no/; (b) Svalbard Integrated
100 Arctic Earth Observing Systems SIOS, not publicly available.

*Figure 5. Number of cross-sections that show a certain distance between cliff top and shoreline for 2010 and 2021. The distribution only changes slightly with clusters around 4* m, 12 m and 24 m.

Line 370: In this study, we analyzed the retreat of the cliff top along the lithified coast of Brøgger peninsula. We showed that this rate is representative of the coastal retreat (Sect. 4.2) so that a comparison with the erosion along other coastlines on Svalbard and in the Arctic can be drawn.

110

We performed a more detailed analysis on environmental conditions influencing coastal erosion. This includes a new section in the methods (Sect. 3.5) and results (Sect. 4.4), as well as a more thoroughly discussion (Sect. 5.2). The revised version includes the following changes:

Changes in the methods: 115

**Line 249: 3.5 Analysis of climate conditions**

We analyzed trends in wind speed and changes in the distance to the sea ice edge (potential wave fetch), as these factors control the interaction between wind and water and therefore the wave field (Barnhart et al., 2014), playing an important 120 role along the coastline of Brøgger peninsula due to mechanical abrasion through wave action. The wind speeds records were taken from the Ny-Ålesund climate station SN99910 (78°55'23" N, 11°55'55" E; Fig. A1), covering the time period 1975 to 2020 (Norwegian Centre for Climate Services, 2023). We extracted days during which mean hourly wind speeds of at least  $10.8 \text{ m s}^{-1}$  (strong breeze or stronger) were recorded, corresponding to large waves of approximately 3 to 4 m (NTNU, 2023).

125

130

We also analyzed the distance to the sea ice edge in northwesterly direction (corresponding to the open Fram Strait), which is the potential distance over which waves can build up. The analysis of this potential fetch was based on data provided by the Norwegian Meteorological Institute (2023) for the time period 1997 to 2023. Hereby, we define the sea ice edge as the given category of 10 to 40 % sea ice concentration, following Meier and Stroeve (2008) and Overeem et al. (2011), who applied a threshold of 15 %. We determined the distance to the sea ice edge for September for which 22 ice charts per year were available on average. For trend detection, we applied a Bayesian regression analysis (Särkkä, 2013), which is explained in Appendix A. In all other cardinal directions, land is found in about 10 to 15 km distance, so that the potential fetch is limited.

In addition, we used hourly ERA5 reanalysis data (Hersbach et al., 2020) in conjunction with the topography-based downscaling routing TopoSCALE (Fiddes and Gruber, 2014) to analyze trends in mean annual air temperature, annual rain- and 135 snowfall, as well as mean annual incoming longwave and shortwave radiation. The methods and detailed results for these climatic parameters are presented in Appendix A and Appendix B.

Changes in the results:

140

**Line 349: 4.5 Changes in climatic conditions**

Given the lithology along the coastline of Brøgger peninsula, mechanical abrasion through wave action is likely a dominant factor for erosion. Therefore, we focus in this section on the wind conditions and changes in the sea ice cover. Other factors,
such as precipitation patterns, air temperature and radiation, are presented in Appendix B.

The trend analysis of wind speeds in Ny-Ålesund defined as a strong breeze or stronger (wind speeds >= 10.8 m s-1, corresponding to large waves of approximately 3 to 4 m; NTNU, 2023) shows an increase by 5.4 days per decade from approximately 65 days on average in the 1970s to 90 days in the last decade (Fig. 8a). However, due to the strong variability, the evidence in favor of a trend remains weak (Bayes factor

---

## Referee Report (RR1)

The authors considerably modified the manuscript to address suggestions by all three reviewers. The study is placed in a wider context and the environmental factors are explored in more detail. I only have few comments (lines refer to the clean version 3 of the manuscript):

- Line 28: What was the time period for the Canadian Beaufort Sea study that gave different rates from those in the database?
- Line 69: I would suggest re-phrase this sentence. Many rock coasts were shown no to respond directly to drivers (storms) and often the response is lagged (e.g. Trenhaile, 1987). However, what the short-term studies weak side is (compared to the longer term studies) is the fact that they may not include high-magnitude low-frequency cliff failure events (under-estimation of 'general' erosion rates) or, conversely, include them (over-estimation of the rates). This makes extrapolating erosion to longer periods and predicting future uncertain.
- Line 72: What changes do you refer to as 'these changes'?
- Line 252: This comment is also relevant to a few other lines listed below. Abrasion is one of the mechanisms of bedrock erosion by waves which requires a tool (such as sand) to hit the rock with. Pneumatic and hydraulic wave action also cause erosion by waves. Therefore, I urge you not to use the term 'mechanical abrasion through wave action' or similar, but refer to it as 'erosion by waves' or 'wave action' depending on the context.
- Figure 5: Provide the total number of cross-sections in the plot or caption.
- Figure 6: Colour scheme is hard to follow. I suggest using a single colour scale starting from white at the lowest rate and increasing darkness. Alternatively blue-to-red will work better than green-to-blue.
- Line 312: It is subject to erosion above threshold of detection.
- Line 350: see comment to Line 252
- Line 379: see comment to Line 252
- Line 383: see comment to Line 252
- Line 384: Do you mean weathering by decreasing resistance to erosion?
- Line 390: Add 'Hornsund' after 'Veslebogen'.
- Line 391: A 100 m transect where?
- Line 397: In space AND time rather than IN?
- Line 404-6: I am not sure if this is a satisfactory reason for higher rates than elsewhere. Be more specific – higher rates may be partly caused by the polar location where in situ bedrock disintegration occurs due to the frost weathering, increasing temperatures contribute to bedrock weakening (permafrost thawing, change of efficiency of thaw/freeze cycle) and increasing exposure to larger waves (effect of storminess increase and sea ice decline).
- Line 419: see comment to Line 252. Perhaps 'wave action and thermo-abrasion'.
- Line 422: see comment to Line 252. Perhaps 'erosion by wind-generated waves'.
- Line 438: Somewhere (perhaps earlier) you should mention that the two metrics you use – distance from the sea ice for the fetch and the wind speed – are not the only ones that play a role in wave action at the Svalbard coasts. The other determinants on the coastal wave energy are for example waves generated on the Greenland Sea far from the study area that arrive there as a long oceanic swell regardless of the local wind conditions, bathymetry and characteristics of the sea ice which impact wave attenuation (density, extent, floe size…). As you do not have direct wave measurements, the two metrics you use may be a good enough.
- Line 440: see comment to Line 252.
- Line 497: Put in brackets the timespan and coastal length.

---

## Referee Report (RR2)

Second review of "Coastal retreat rates of high-Arctic rock cliffs on Brøgger peninsula, Svalbard, accelerate during the past decade" by Aga et al. [Preprint egusphere-2023-321], submitted to Esurf

**General comments**

I express my gratitude to the authors for their dedicated efforts in enhancing this manuscript and for their responsiveness to the reviewer's feedback. While commendable progress has been made overall, certain concerns persist.

The manuscript has been significantly expanded by the authors, contributing to a better understanding of their procedures. However, in specific instances, particularly in the introduction, the additional information extends beyond the study's scope, needlessly elongating the paper without offering substantial benefits. Some sections exhibit an overly expansive writing style, and adopting a more concise approach would enhance overall readability. A comprehensive analysis of the study area and methods would be beneficial to discern which phenomena and methods are pertinent to the paper's main objectives and which are solely relevant for study reproducibility. Consider relocating sections 3.3 and 4.2 to the appendix, accompanied by a short statement in the main text outlining the actions taken and directing readers to the appendix for further details.

In the discussion (chapter 5.1), the authors draw comparisons with previous studies and discuss potential differences. However, this study is not a review, and comparing it to previous research is not its primary objective. The first half of chapter 5.1 can be summarized by stating that the measured retreat rates align with previous studies (+citations). The latter half of the chapter appears speculative without clear direction. The initial discussion chapter should focus on the main results, addressing shortcomings and uncertainties. Additionally, the authors should acknowledge the higher uncertainty in cliff top position compared to measured retreat and clarify how they ensure the measurement reflects actual retreat rather than uncertainty.

While the authors rightfully emphasize the significance of long time-series analysis, especially in the Arctic's high interannual variabilities, the presentation of a one-year dataset of rocksurface temperatures appears incongruent. Comparing decadal analysis of cliff-top retreat rates with a one-year rock surface temperature dataset may not be appropriate, and the latter adds minimal value to the study. If inclusion is necessary, relocating this dataset to the appendix is recommended, although removal is preferable.

Line numbers below refer to the revised manuscript with tracked changes.

**Abstract**

Line 9: Consider dividing this lengthy sentence regarding the retreat rates of the north and south-facing coastlines into two sentences for improved readability. Could you also clarify whether you are reporting the standard deviation or the measurement uncertainty (DOA) for each decade? Notably, for the northeast-facing coast, the uncertainty range consistently equals or surpasses the measured change in each decade. This prompts the question of how you can ascertain the presence of any statistically significant change in these instances.

**Introduction**

Line 23: The inclusion of pan-Arctic erosion rates or rates from Siberia and Alaska in this paper seems irrelevant, given the substantial differences in erosion rates and study areas compared to the site under investigation. It would be more beneficial to omit this added information and instead concentrate on erosion rates from studies that are more comparable in terms of landscapes and methodologies.

Line 40: While Lantuit et al. present values for Nunavut, the basis for these rates is not clearly stated in their paper. In my view, the most noteworthy aspect of their study concerning lithified coasts is the revelation of the considerable gaps in our understanding of Arctic coastal dynamics.

Line 52: What accounts for higher erosion rates on bedrock cliffs covered by unconsolidated sediment, and why are these rates not observed at your study site?

Line 61: Kindly replace "bluff" with "cliff."

**Study area**

Line 81: Could you provide more detailed information about the beach, such as whether it is a mixed sand and gravel beach, and if there are variations in sediments between the northeast and southwest facing sections? Additionally, a more comprehensive description of the coastal setting, including details on tidal range, fetch, and beach morphology, would enhance clarity.

Line 86: The current wording of this sentence suggests that the raised beach ridges were situated at 45 m.a.s.l. during the Late Weichsel and have been uplifted since then. Could you please clarify this point?

Lines 97 to 105: The discussion of potential driving factors for coastal cliff erosion would be more fitting in the overall discussion section, especially when considering how these factors might change with climate change.

Line 106: "Weather station" was correct; kindly revert the change.

Lines 106 to 117: Please consider condensing this paragraph significantly, focusing only on information directly relevant to your objectives.

Line 114: Replace "found" with "measured."

Line 129: Remove "have been recorded."

Line 131: If the information is not applicable, consider removing it.

**Data and methods**

Line 164: Could you provide details on the specific methods employed to verify the accuracy of the geo-referencing?

Lines 174-179: Rather than delving into the distinctions between RTK and PPK, a concise statement mentioning that you corrected points using PPK and specifying the associated uncertainty would suffice. Additional details can either be omitted or moved to the appendix.

Line 194: While it's commendable that you explained the calculation of cliff top retreat rates and introduced a proxy for the relationship between cliff top and coastline retreat, it would be beneficial to include information on the local geo-referencing error for the orthoimages used in distance measurements. To enhance local accuracy, consider either geo-referencing one image to another using a method like spline transformation or including a control in the provided data by measuring distances between two stable objects near the cliff in both images and comparing the results.

Line 194: It seems you measured planar distance. Could you please clarify?

Line 195: How have you factored in wave run-up and tides? Figure 4b suggests that wave run-up may play a significant role at your study site when determining the shoreline's position.

**Results**
Figure 4: Why is there a zero next to the cross-section?

**Discussion**
Line 401: A comparison with a panarctic retreat rate may be unsuitable for a case study of this nature.

Line 406: The reference to Lantuit et al. 2012 does not appear to contain original data. Please consider citing the original Nunavut study if available or omitting this sentence.

Line 407: Is it appropriate to compare potential driving factors at your local site with those of the entire Canadian archipelago? The scale and spatial homogeneity might present challenges for a meaningful comparison.

Line 451: Could there be an icefoot developing in front of the cliff? If so, do these cyclones influence the retreat rates, or does the icefoot protect the cliff?

Line 480: I would suggest adding 'in the next decades' or 'until the end of the century.' In the long term, the relative sea level fall is expected to increase more than coastal retreat, leading to advancing coastlines.

Lines 515 and following: I respectfully differ with my colleague regarding the discussion of implications for infrastructure at this point. This is neither an objective of the study nor relevant, as there is no infrastructure at risk at your study site. Furthermore, there is no novel data presented on infrastructure at risk that warrants discussion here. It would be more pertinent to focus on your data in the discussion without turning it into a review of other studies.

**Data availability**

I strongly urge the authors to ensure all data is made accessible. Adhering to FAIR (findable, accessible, interoperable, reusable) data practices is highly recommended. Additionally, please provide comprehensive metadata for the uploaded data. While attempting to comprehend the "Distance_multiple_digitization_of_coastline" file, I encountered difficulty deciphering its contents.

---

## Author Response (AR3)

**Letter to the editor**

We would like to thank the editor for handling our manuscript and accepting it for final publication in ESurf. We made minor revisions to the wording of the text to improve clarity as suggested by the editor. We attached the track-changes document in the following.

[revised manuscript text omitted]